# Aquatic and Soil CO₂ Emissions from forested wetlands of Congo's Cuvette Centrale

Antoine de Clippele[1*], Astrid C. H. Jaeger[1*], Simon Baumgartner[2,3], Marijn Bauters[4], Pascal Boeckx[5], Clement Botefa[6], Glenn Bush[7], Jessica Carilli[1], Travis W. Drake[1], Christian Ekamba[8], Gode Lompoko[6], Nivens Bey Mukwiele[6], Kristof Van Oost[3], Roland A. Werner[1], Joseph Zambo[7], Johan Six[1], Matti Barthel[1]

1 Department of Environmental Systems Science, ETH Zurich, Switzerland

2 Research Division Agroecology and Environment, Agroscope, Zurich, Switzerland

3 Earth and Life Institute, Université Catholique de Louvain, Louvain-la-Neuve, Belgium

4 Q-ForestLab, Department of Environment, Ghent University, Belgium

5 Isotope Bioscience Laboratory, Department of Green Chemistry and Technology, Ghent University, Belgium

6 ICCN Jardin de Botanique d'Eala, Mbandaka, Democratic Republic of Congo

7 Woodwell Climate Research Center, USA

8 Coordination Provinciale de l'environnement, Mbandaka, Democratic Republic of Congo

*These authors contributed equally to this study.

*Correspondence to* : Antoine de Clippele (antoine.declippele@usys.ethz.ch)

**Abstract.** Within tropical forest ecosystems, wetlands such as swamp forests are an important interface between the terrestrial and aquatic landscape. Despite this assumed importance, there is a paucity of carbon flux data from wetlands in tropical Africa. Therefore, the magnitude and source of carbon dioxide ($CO_2$) fluxes, carbon isotopic ratios, and environmental conditions were measured for three years between 2019 to 2022 in a seasonally flooded forest and a perennially flooded forest in the *Cuvette Centrale* of the Congo Basin. The mean surface fluxes for the seasonally flooded site and the perennially flooded site were $2.36 \pm 0.51$ µmol m$^{-2}$ s$^{-1}$ and $4.38 \pm 0.64$ µmol m$^{-2}$ s$^{-1}$ respectively. The time series data revealed no marked seasonal pattern in $CO_2$ fluxes. As for the environmental drivers, the fluxes at the seasonally flooded site exhibited a positive correlation with soil temperature and soil moisture. Additionally, the water level appeared to be a significant factor, demonstrating a quadratic relationship with the soil fluxes at the seasonally flooded site. $\delta^{13}C$ values showed a progressive increase across the carbon pools, from above-ground biomass, then leaf litter, to soil organic carbon (SOC). However, there was no significant difference in $\delta^{13}C$ enrichment between SOC and soil respired $CO_2$. This lack of enrichment can be attributed to either a significant contribution from the autotrophic component of soil respiration or a result of closed system dynamics.

An *in-situ* derived gas transfer velocity ($k_{600} = 2.95$ cm h$^{-1}$) was used to calculate the aquatic $CO_2$ fluxes at the perennially flooded site. Despite the low $k_{600}$, relatively high $CO_2$ surface fluxes were found due to very high partial pressure of $CO_2$ ($pCO_2$) values measured in the flooding waters. Overall, these results offer a quantification of the $CO_2$ fluxes from forested wetlands and provide an insight of the temporal variability of these fluxes as well as their sensitivity to environmental drivers.

# 1 Introduction

Along with the oceans and Northern Hemisphere forests, tropical forests represent one of the three main components of the global carbon sink (Mitchard, 2018). However, due to relatively high gross primary productivity, temperature and soil moisture, tropical forest soils also constitute a large terrestrial source of carbon dioxide ($CO_2$). Indeed, tropical regions are estimated to contribute up to 64% of global soil respiration, rendering it as the largest flux of $CO_2$ from terrestrial ecosystems to the atmosphere (Hashimoto et al., 2015; Huang et al., 2020).


Wetland cover in the tropical Congo Basin is estimated to range between 332,620 and 359,556 km² (Bwangoy et al. 2010; Fatras et al. 2021). This area includes the *Cuvette Centrale*, which spans approximately 167,600 km² and hosts lowland and swamp forests, including the largest peatland complex across the tropics (Crezee et al., 2022). With catchment drainage from north and south of the equator as well as sustained rainfall at the center of the basin (Breitengroß, 1972; Runge, 2007), the

*Cuvette Centrale* shows near permanent inundation. Characterizing $CO_2$ fluxes in this extensive region is especially important since inland waters are increasingly recognized as significant sources of greenhouse gases (GHG) within the terrestrial landscape (Bastviken et al. 2011; Drake, Raymond, and Spencer 2018; Borges, Darchambeau, et al. 2015; Rosentreter et al. 2021) and notably in global carbon dioxide emissions (Raymond et al. 2013). Recent data additionally suggests that the Congo Basin's inland waters might emit more carbon (C) per area than their counterparts in the Amazon Basin (Alsdorf et al., 2016).

Profound hydrological (Alsdorf et al., 2016), structural (Lewis et al., 2013), ecological (Parmentier et al., 2007; Slik et al., 2015), aquatic biogeochemical (Borges et al. 2015), and terrestrial biogeochemical (Hubau et al., 2020) differences indicate that GHG flux estimates cannot simply be transferred from the Neotropics to the Afro-tropics. However, while recent research on GHG emission from the Congo Basin has focused on either riverine (Borges et al., 2019; Bouillon et al., 2012; Mann et al., 2014a; Upstill-Goddard et al., 2017) or terrestrial fluxes (Baumgartner et al. 2020; Gallarotti et al. 2021; Barthel et al. 2022;

Daelman et al. 2024), direct measurements from forested wetlands are still lacking. Despite its immense global importance, only two studies, to the best of our knowledge, have been looking into GHG emissions from Congo's wetlands (Tathy et al. 1992; Barthel et al. 2022).

Forested wetlands/swamp forests are located at the transition zone between the terrestrial and the aquatic realm. The duration

and seasonality of the flooding in the forests will constrain the contribution from/to the river system. While flooded, the swamp forests are connected to the river system and receive and/or discharge materials from/to the river network (Aufdenkampe et al., 2011). Variations of riverine greenhouse gas concentrations have been shown to be driven by fluvial-wetland connectivity for the *Cuvette Centrale* based on data from 10 expeditions across the Congo River network (Borges et al., 2019). Furthermore, streams and rivers draining Congo's flooded forests were found to have the highest dissolved concentrations of $CO_2$ among

different land cover types in the Basin, indicating the substantial contribution of forested wetlands on the overall inland water GHG budget (Mann et al., 2014b).

Here, we report three years of carbon dioxide ($CO_2$) fluxes measured from two sites situated within the *Cuvette Centrale*: a seasonally flooded forest site and a perennially flooded forest site. During the observation period, surface $CO_2$ fluxes whether soil or aquatic, were measured fortnightly to capture the seasonal and inter-annual variation of the fluxes. Hence, these results provide insights into the temporal dynamics of $CO_2$ fluxes in forested wetlands across two different flooding regimes.

## 2 Materials and Methods

### 2.1 Study sites

The sites were located near the town of Mbandaka (Democratic Republic of the Congo, Équateur province), which is located at the Ruki-Congo confluence within the *Cuvette Centrale* (Figure 2). The mean annual precipitation and mean annual temperature of the sampling area are 1588 mm and 25 ℃, respectively (see the measurements detailed below). The long dry season in Mbandaka typically lasts from July to August, while the short dry season occurs between January and February. Here, surface $CO_2$ fluxes were measured at two different sites across two different hydrological regimes, one in a seasonally flooded forest (N 0.06335, E 18.31054, 300 m a.s.l.) – referred to as SFF site –and in a perennially flooded forest (S 0.03135, E 18.3102, 305 m a.s.l.) – referred to as PFF site (Figure 1).

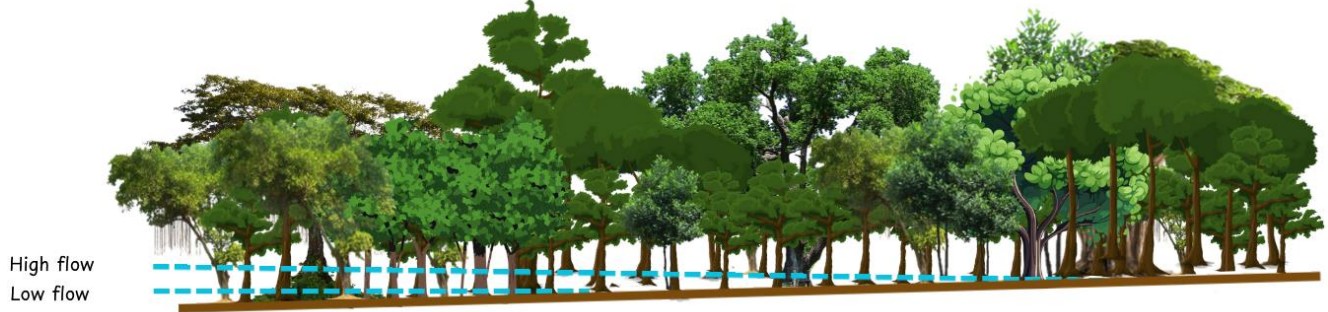

High flow
Low flow

Perennially flooded
forest site (PFF)

Seasonally flooded
forest site (SFF)

**Figure 1. Diagram showing the location of the two experimental sites (PFF and SFF) relative to the hydrological gradient.**

The seasonally flooded forest site (SFF) investigated was located within a botanical garden 7 km from the center of Mbandaka (*Jardin Botanique d'Eala,* operated by the *Institute Congolais pour la Conservation de la Nature* (ICCN)). The botanical garden comprises 371 ha of land consisting of 35% dense swamp forest, 14% forest on firm ground, 32% open forest, and the remaining area consisting of secondary forest, grassland, and deforested land, of which 189 ha are protected forest area. There are 3500 different trees and herbaceous plant species, with the main tree species being *Hevea brasiliensis, Ouratea arnoldiana, Pentaclethra eetveldeana, Strombosia tetandra*, and *Daniella pynaertii*. The soil at the site, covered by a thick litter layer, was characterized as Eutric Gleysols (texture 42/50/8 sand/silt/clay in %, bulk density 1.27 g cm$^{-3}$). The litter layer harbors a dense

mesh of fine roots, whereas almost no roots were found to penetrate the upper mineral soil layer (0-30 cm). The SFF site is
seasonally flooded from about December to January (~2 months).

At the SFF site, combined soil moisture and temperature sensors (ECH$_2$O 5TM, Meter Group, Inc. USA) connected to loggers
(Em50, Meter Group, Inc., USA) were installed at 10 and 30 cm depth, respectively. The data was recorded every 6 h.
Unfortunately, one logger was stolen and the other logger stopped working during deployment; thus, data is only available
from November 2019 to July 2020 (Figure 3). Afterward, TMS-4 dataloggers (TOMST, Czechia) were installed in December
2020 to record surface volumetric soil water content (0-14cm) and soil temperature at 8 cm depth in 15-minute intervals. Raw
data (soil moisture count) retrieved from TMS-4 dataloggers was converted into soil VWC with calibrations curves, following
Wild et al. (2019), using site-specific soil properties (soil texture: 42/50/8 sand/silt/clay in %, bulk density 1.27 g cm$^{-3}$) and
measured soil temperatures. The soil VWC values from the ECH$_2$O 5TM sensors showed a systematic offset compared to
those obtained from the TMS-4 dataloggers. This was attributed to instrument artefact and corrected by using the difference
between maximum values. Furthermore, precipitation, air temperature, relative humidity, solar radiation, and wind speed data
were retrieved for the observation period from the Trans-African Hydro-Meteorological Monitoring Observatory (TAHMO)
station located in close vicinity to the forest site (ATMOS 41, Meter Group, Inc., USA).

The perennially flooded forest (PFF) site is located about 8 km upstream of the Congo-Ruki confluence, following a small
side tributary named Lolifa. The headwater stream area is completely flooded for most of the year, making the stream bed
channel indistinguishable. This creates a continuous wetland area where the PFF site is located. While the water is mostly
stagnant at the site, a small drainage flow appears during the dry season (late June to early September). The site was accessed
with a motorized dugout canoe, and sampling was done fortnightly from the side of the canoe. The main tree species at the
PFF site were *Uapaca sp, Irvigia smitii, and Daniella pynaertii De Wild*. Additional to the surface CO$_2$ fluxes, water samples
were collected on the same day to measure pH, dissolved organic carbon (DOC) and total dissolved nitrogen (TDN). The
presented C:N ratio was thus calculated using TDN rather than dissolved organic nitrogen (DON). Previous analyses (Drake
et al., 2023) showed that TDN consistently comprised an average of 90% of DON and thus reflected well the relative changes
of DON concentrations. The specific methods used for sample processing and analysis as well as the calculations are described
in Drake et al., (2023).

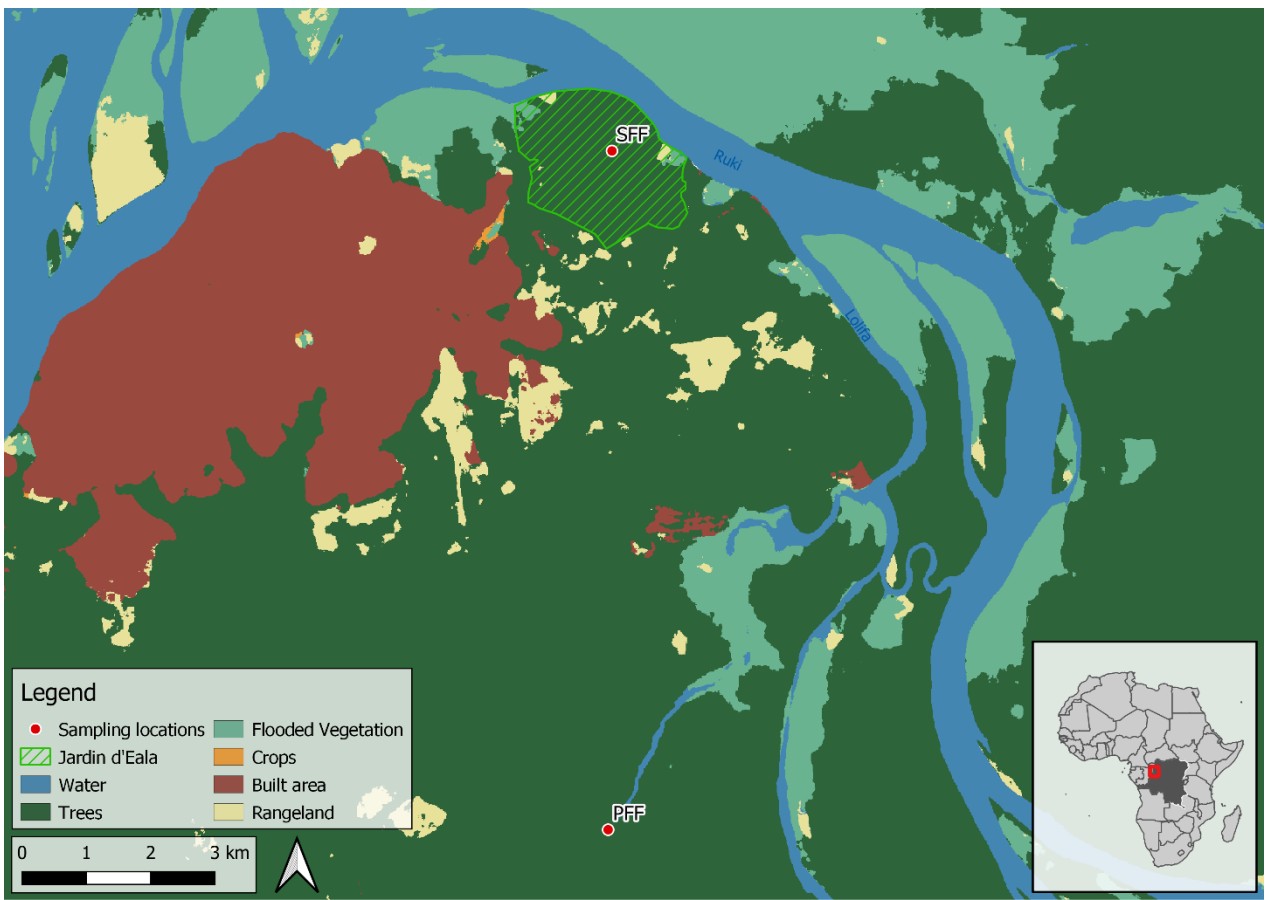

Figure 2. Map presenting the two sampling sites in the vicinity of Mbandaka (Democratic Republic of the Congo). The boundaries of the Jardin botanique d'Eala are highlighted. Map data: © 2020-2023 Impact Observatory, Inc. and GADM.

## 2.2 CO$_2$ fluxes

### 2.2.1 CO$_2$ surface fluxes at the SFF site

A total of six polyvinylchloride soil flux chambers (height = 0.3m, diameter = 0.3m) were installed in November 2019 at the SFF site. The SFF site was chosen as a representative site of the surrounding forest. The six chambers were spaced about 20 meters apart, randomly distributed across the site but accounting for variations in local microtopography. The chamber bases were inserted approximately 5 cm into the ground and remained in place throughout the measurement period, with the chambers left open except during sampling. Chambers affected by seasonal flooding were measured for as long they were not completely submerged at which point floating chambers (V = 17L) were used instead. Sampling with the soil chambers was conducted fortnightly for three consecutive years (2019/11 – 2022/12), totalling 403 flux measurements. Sampling was interrupted once for about six months due to logistical constraints (first half of 2022).

Each chamber lid was equipped with a thermocouple to measure headspace temperature, a vent tube to avoid pressure changes, and a sampling port. The sampling port had a 3-way Luer valve attached, connecting the syringe, needle, and chamber. Before withdrawing each gas sample from the headspace, chamber air was mixed by moving the syringe plunger several times; for soil GHG flux determination, gas samples were taken at timesteps of 20 min throughout 1 hour (t1 = 0 min, t2 = 20 min, t3 = 40 min, t4 = 60 min). A longer chamber closure time than recommended (Pavelka et al., 2018) was used to obtain robust Keeling plots along with the flux measurements. At each timestep, 20 mL of gas sample were stored in 12 mL pre-evacuated vials (Labco, UK) using a gas-tight disposable plastic syringe (20 mL). The resulting vial overpressure prevents air ingress due to temperature and pressure changes potentially occurring during transport and is required for sample withdrawal by the GC autosampler. To aid vacuum and sample preservation, each evacuated vial was sealed with an additional silicone layer (Dow Corning 734, Dow Silicones Corporation, USA). Soil $CO_2$ fluxes were calculated via linear concentration increase over time using the ideal gas law $PV = nRT$:

$$n = \frac{PV}{RT} \ (1) \quad \text{and} \quad F = \frac{\Delta n}{\Delta t} S^{-1} \ (2)$$

with $n$ moles of gas [mol], $P$ partial pressure of trace gas [atm µmol mol$^{-1}$], $R$ gas constant 0.08206 [L atm K$^{-1}$ mol$^{-1}$], $T$ headspace temperature [K], $F$ flux of gas [µmol m$^{-2}$ s$^{-1}$], $\frac{\Delta n}{\Delta t}$ rate of change in concentration [mol s$^{-1}$], $V$ chamber volume [L], and $S$ surface area enclosed by chamber [m$^2$]. The coefficient of determination ($r^2$) for the linear regression of $CO_2$ yielded a $r^2 > 0.95$ for 95% of the data (Supp. Fig. 1). All data with $r^2 > 0.1$ was kept for the statistical analyses. Such a low $r^2$ threshold was maintained because fluxes with low $r^2$ values are typically related to low flux rates rather than due to methodological or technical issues. Increasing the threshold would introduce a bias toward higher fluxes in the data.

**2.2.2 Aquatic surface fluxes at the PFF site**

The aquatic surface flux to the atmosphere ($F_{CO_2}$, µmol m$^{-2}$ s$^{-1}$) from the PFF site was estimated according to a simple gas transfer model (Mann et al. 2014):

$$F_{CO_2} = k_x \cdot K_H \cdot (pCO_{2w} - pCO_{2a}) \ (3)$$

where $k_x$ is the freshwater gas transfer velocity of $CO_2$ [m s$^{-1}$], $K_H$ is the Henry's constant for $CO_2$ [mol m$^{-3}$ atm$^{-1}$], $pCO_{2w,a}$ the partial pressure of $CO_2$ in water and atmosphere, respectively [µatm].

Since the magnitude of the gas transfer velocity is governed by numerous factors (e.g., wind speed, water current velocity, slope), an *in-situ* gas transfer velocity $k$ was calculated as 3.5 cm h$^{-1}$ using the aquatic fluxes from the SFF site sampled between 2022-07 to 2022-12 with the above-mentioned floating chamber (V = 17 L) and the corresponding dissolved $CO_2$ concentrations of the inundation water at the same site. The value of 3.5 cm h$^{-1}$ was then applied to the perennially flooded

forest dataset where no floating chamber measurements existed. Hence, fluxes from the PFF site were derived using the measured gas transfer velocity from the SFF site (3.5 cm h$^{-1}$).

In order to compare the *in-situ* derived velocity $k_x$ with the temperature normalized transfer velocity ($k_{600}$) for tropical wetlands of 2.4 cm h$^{-1}$ (Aufdenkampe et al., 2011), we used the equation from Pelletier et al. (2014) to convert $k_x$ to $k_{600}$.

$$k_x = k_{600} \left(\frac{S_c}{600}\right)^{-b} \quad (4)$$

Where $S_c$ is the gas specific Schmid number and b derived from literature (0.66 for wind speed $\leq$ 3 m/s; Pelletier et al. 2014). The gas specific Schmid number is a function of water temperature (T in °C) as defined by Wanninkhof (2014):

$$S_{cCO_2} = 1923.6 - 125.06T + 4.3773T^2 - 0.085681T^3 + 0.00070284T^4 \quad (5)$$

For $pCO_{2a}$, the tropospheric mean value from the year 2020 (400 µatm) was used while $pCO_{2w}$ was determined using the headspace equilibration technique. That is, 6 mL of bubble-free water sample were injected with a syringe into a 12 mL N$_2$-pre-flushed vial (Exetainer®, Labco, UK) pre-poisoned with 50 µL of 50% ZnCl$_2$ to stop the microbial activity. After sufficient equilibration time, the remaining headspace was analysed for $CO_2$ concentrations using a gas chromatograph (see Section below), and total dissolved concentrations were calculated based on Henry's law (see a detailed method in Supplementary). For each date, $pCO_{2w}$ samples were taken in triplicate with an average coefficient of variation (CV) of 8%.

### 2.3 Gas Chromatography

Gas samples were analyzed at ETH Zurich using a gas chromatograph (GC; Bruker, 456-GC, Scion Instruments, Livingston, UK) separating $CO_2$ from residual air. After separation, the concentration of $CO_2$ was measured on a thermal conductivity detector. GC calibration was done with a suite of three standards (Carbagas AG, Switzerland; PanGas AG, Switzerland) across a concentration range from 249 to 3040 ppm $CO_2$. Each standard was analysed ten times at start, middle, and end of each set of 140-180 samples. Moreover, because of occasional high $CO_2$ sample concentrations, an entire system flush was done between each sample measurement to avoid any carry-over effects. The same GC setup was used for both flux samples and dissolved $CO_2$ samples.

### 2.4 δ$^{13}$C of soil-derived $CO_2$ fluxes and dissolved $CO_2$

The carbon isotopic composition of the $CO_2$ samples was analysed for one SFF $CO_2$ flux sample set of each month. That is, after $CO_2$ concentration measurement with the GC, the same samples were analysed for δ$^{13}$C of $CO_2$ with a modified Gasbench II periphery (Finnigan MAT, Bremen, D) coupled to an isotope ratio mass spectrometer (IRMS; Delta$^{plus}$XP; Finnigan MAT) as described in Baumgartner et al. (2020). Post-run off-line calculation and drift correction for assigning the final δ$^{13}$C values

on the Vienna Peedee Belemnite (V-PDB) scale was done following the 'IT principle' (Werner and Brand, 2001). The $\delta^{13}$C-values of the laboratory air standards were determined at the Max-Planck-Institute for Biogeochemistry (Jena, Germany), according to Werner, Rothe, and Brand (2001). The final soil $CO_2$-$\delta^{13}$C values were calculated using the Keeling plot approach (Keeling, 1958) (Supp. Fig. 2).

$\delta^{13}$C of dissolved riverine $CO_2$ was determined using the headspace equilibration technique as described in section 2.2.2. Instead of concentration, $\delta^{13}$C of the headspace was analysed via IRMS as described above. Samples were taken each month from the Ruki river between 10-2022 to 06-2023 with 2-3 replicates per sampling (Supp. Fig. 4).

## 2.5 $\delta^{13}$C of leaves, litter, and soils

Fresh leaf samples were taken from a range of the most representative tree species at two different timepoints (2019-11; 2023-11). In addition, litter samples were collected at the same time and both were used to analyze the carbon isotopic composition ($\delta^{13}$C). Before analysis samples were dried, homogenized, and ground. Soil samples were taken in 2019-11, 2020-02 and 2023-11 at 0-30 cm depth and air dried, sieved, and milled. All samples were analyzed using an elemental analyzer (Flash EA 1112 Series, Thermo Italy, former CE Instruments, Rhodano, Italy), interfaced with an IRMS (Finnigan MAT Delta[plus] XP, Bremen, Germany) via a 6-port valve (Brooks et al., 2003) and a ConFlo III ( Werner et al. 1999). Soil samples are subsequently referred to as soil organic carbon (SOC) samples. Calibration of laboratory standards (acetanilide, caffeine, tyrosine) was done by comparison to the corresponding international reference materials provided by the IAEA (Vienna, Austria).

## 2.6 Water level

Direct measurements of the water level were not available for the whole observation period. Previous work showed a linear relationship between the water level of the Congo River and the Ruki river (unpublished, Supp. Fig. 3). Additionally, the rainfall and/or the hydrological dynamics of the river influence the water levels in the wetlands. In the *Cuvette Centrale*, Georgiou et al. (2023) determined that the water levels of riverine locations in the Democratic Republic of the Congo (DRC) correlate more with the hydrological dynamics of the river system than with the rainfall input. Hence, available daily measurements of the water level of the Congo River in Mbandaka were used as a proxy of the water level below- and aboveground at the SFF site. (Supp. Fig. 6). This data was extracted from an almost continuous record of water gauge readings, collected in vicinity of the SFF site (~4 km) by the Congolese public institution, *Régie des Voies Fluviales*, since 1913.

## 2.7 Statistical Analyses

Daily environmental conditions were used to explain variability in the measured soil $CO_2$ fluxes (n = 403) at the SFF site. For this, a linear mixed effects model was fitted using soil temperature, volumetric soil water content, and river level as fixed effects. River level showed a non-linear relationship with surface fluxes. Hence, a quadratic term was added to account for the

non-linear effect. The predictor variables were standardized before fitting the models. All models controlled for repeated measurements in the same chambers, by adding chamber ID as a random intercept. Models were fitted by the restricted maximum likelihood method using *lme4* (Bates et al., 2015). Full and reduced models were compared using likelihood ratio test and adjusted $r^2$ values using *MuMin* package (Barton, 2020). Furthermore, a backward stepwise regression analysis was conducted on the full model, incorporating all effects and interaction terms, to identify the most parsimonious model with

highest explanatory power (Kuznetsova et al., 2017). The resulting model included an additional interaction term between soil moisture and river level. However, this term was subsequently removed due to multicollinearity and its lack of practical significance. Marginal and conditional $r^2$ values for mixed effects were calculated using Nakagawa et al., (2017), inclusive $r^2$ estimated with partR2 package (Stoffel et al., 2021)  and p-values using Satterthwaite's approximation with the *lmerTest* package (Kuznetsova et al., 2017). Additionally, confidence intervals for the effect estimates were computed to confirm the

interpretation of the estimated parameters. The assumptions of the model were validated by verifying the linearity, normality and homoscedasticity of the residuals. Multicollinearity between the predictor variables was also estimated (Variance Inflation Factor (VIF) inferior to 3). Statistical differences between $\delta^{13}C$ values measured across the different carbon pools were tested with the Kruskal-Wallis test, followed by a pairwise Wilcoxon comparisons. Significance was established when the Bonferroni adjusted p-values were inferior to 0.05. Statistical and graphical data analysis were done in R v.4.3.2 (R Core Team, 2023) via

RStudio v.2023.12.0 (RStudio Team, 2023), using the packages *tidyverse* v.2.0.0, *tydr* v1.3.0, *dplyr* v.1.1.4 (Wickham et al., 2023), *ggplot2* v. 3.4.4 (Wickham, 2009), sjPlot (Lüdecke, 2013) and *lubridate* v.1.9.3 (Grolemund and Wickham, 2011). QGIS version 3.16 was used to compile the map of the sampling locations.

## 3 Results

### 3.1 Environmental Conditions

The long dry season in Mbandaka is considered from July to August whereas the short dry season spans between January and February. However, frequent rainfall as shown in Figure 3 renders the region as relatively wet throughout the entire year. Annual precipitation was the highest in 2020 with 1855 mm and lowest in 2022 with 1417 mm (self-measured; Figure 3). The flooding period at the study site is typically centered around December and January. The highest weekly precipitation occurred in July and September of each year with 120 – 182 mm (Figure 3A). Overall, the weekly precipitation ranged from 0 – 182

250  mm, with a monthly average of 31mm (Figure 3A).

Volumetric soil water content, hereinafter referred to as soil moisture, averaged $0.60 \pm 0.09$ m$^3$ m$^{-3}$, ranging between 0.35 to 0.76 m$^3$ m$^{-3}$ for the observation period (Figure 3A). In general, the soil moisture showed strong seasonality, with an increase in November and peak values observed in January. Thereafter, the soil moisture decreased before stabilizing until the following

wet season. This pattern was less pronounced over the 2021-2022 season (Figure 3A).

Soil and air temperatures were stable throughout the observation period (Figure 3B). Recorded mean air temperature at the weather station was 25.0 °C (± 0.7 °C), and mean soil temperature at the SFF site was 24.7 °C (± 0.3°C) for the observation period.

**3.2 Soil and aquatic $CO_2$ fluxes**

Over the observation period, $CO_2$ fluxes from the PFF site were higher than from the SFF site (Figure 3F). At both sites, $CO_2$ fluxes exhibited intra-annual variability. However, distinct seasonal patterns were not clear. Notably, at the SFF site, the onset of flooding appeared to induce a decline in fluxes. Furthermore, among the environmental variables, $CO_2$ fluxes exhibited significant correlations with soil moisture, soil temperature, and river level (Table 1). At the PFF site, the highest fluxes were recorded in June and August of 2020 with 5.71 and 5.76 $\mu$mol m$^{-2}$ s$^{-1}$, whereas the lowest values were observed in September and October 2020 with 3.35 and 3.42 $\mu$mol m$^{-2}$ s$^{-1}$. Mean weekly surface fluxes ($F_{CO2}$) from the PFF site ranged from 3.35 to 5.76 $\mu$mol m$^{-2}$ s$^{-1}$ with an average flux of 4.38 ± 0.64 $\mu$mol m$^{-2}$ s$^{-1}$ using the *in-situ* derived gas transfer velocity of 3.5 cm h$^{-1}$ (Figure 3E). Mean weekly surface fluxes ($F_{CO2}$) from the SFF site ranged from 0.87 to 3.64 $\mu$mol m$^{-2}$ s$^{-1}$ with an average of 2.36 ± 0.51 $\mu$mol m$^{-2}$ s$^{-1}$. Here, the lowest flux was observed in July 2022 with 0.87 $\mu$mol m$^{-2}$ s$^{-1}$, period corresponding to the lowest soil moisture recorded (0.35 m$^3$ m$^{-3}$), while peaking in May 2020 with 3.64 $\mu$mol m$^{-2}$ s$^{-1}$ (Figure 3E).

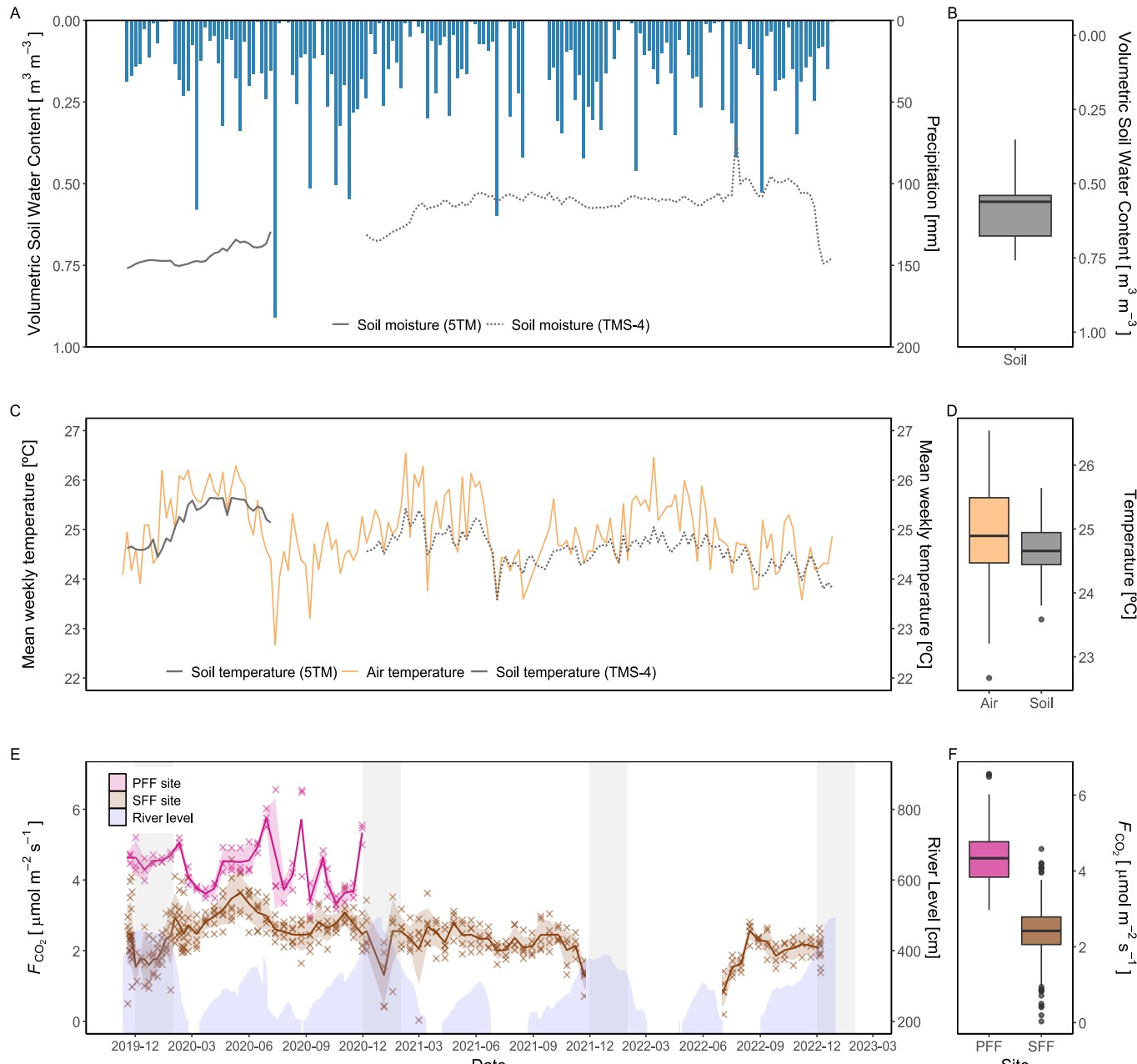

**Figure 3. Weekly precipitation, volumetric soil water content, temperature, and CO₂ fluxes. (A) The sum of the weekly precipitation [mm] (blue) obtained from the Trans-African Hydro-Meteorological Monitoring Observatory and mean volumetric soil water content [m³ m⁻³] measured with soil moisture sensors (ECH₂O 5TM = solid line, TMS-4 dataloggers = dotted line). (B) Distribution of volumetric soil water content [m³ m⁻³], both sensor types combined. (C) Mean weekly air temperature [°C] (gold) was obtained from the Trans-African Hydro-Meteorological Monitoring Observatory. The mean weekly soil temperature [°C] was measured with soil temperature sensors (ECH₂O 5TM = grey solid line, TMS-4 dataloggers = grey dotted line). (D) Distribution of air and soil temperatures [°C], both sensor types combined. (E) Measured surface CO₂ fluxes (cross) [μmol m⁻² s⁻¹] from the SSF site (brown)**

and calculated $CO_2$ fluxes from the PFF site with a K of 3.5 cm h$^{-1}$(pink). Calculated weekly means (line) and the standard error of
280 the mean are displayed. Blue shading represents river levels (see Section 2.6), while grey bands indicate flooding periods (Dec-Jan)
at the SFF site. The displayed time series are discontinuous due to fieldwork constraints (see Section 2.2.) (F) Distribution of surface
$CO_2$ fluxes at the PFF and SFF sites.

### 3.2.1. Controls on surface $CO_2$ fluxes at the SSF site

The linear mixed effects model (n = 324) explained 43.0 % of the total variability, of which 35.4 % is allocated to the fixed
effects (river level, soil moisture and soil temperature; Table 1). The soil temperature and soil moisture are positively correlated
with surface $CO_2$ fluxes. The river level, used as a proxy for the water level, exhibited a quadratic relationship with the $CO_2$
fluxes measured at the SFF site (Table 1; Figure 4C). The nonlinear component exhibited a negative sign, describing an inverse
U-shaped curve (Figure 4C). Initially, the relationship had a positive slope at lower river levels, reaching a maximum point
before transitioning to a negative slope. As the river level is used as a proxy for the on-site water level, a short-term campaign
was done during the wet season 2023-2024 to confirm the influence of the water level with direct measurements (Unpublished;
Supp. Fig. 7). Finally, the significant positive interaction term between temperature and river level suggests a synergistic effect
where the combined influence of these two variables on surface fluxes is greater than the addition of their respective individual
effects (Figure 4D).

**Table 1. Fixed effect estimates for surface $CO_2$ fluxes at SFF site including river level, soil temperature, and soil moisture as standardized predictors, allowing comparison of their relative importance. For each effect, standard error and p-values (Sattherhwaite's method) are estimated, as well as the marginal (m) and conditional (c) $R^2_{adfj}$ (Nakagawa et al., 2017).**

| Response | Effect | Estimate | SE | P-value | $R^2_{adj, m}$ / $R^2_{adj, c}$ |
|---|---|---|---|---|---|
| Surface $CO_2$ flux | Intercept | 2.61 | 0.09 | < 0.001 | 0.354/0.430 |
| | River level [1$^{st}$ degree] | -0.01 | 0.04 | **0.833** | |
| | River level [2$^{nd}$ degree] | -0.19 | 0.04 | < 0.001 | |
| | Soil temperature | 0.18 | 0.04 | < 0.001 | |
| | Soil moisture | 0.28 | 0.05 | < 0.001 | |
| | River level: Soil Temperature[1] | 0.18 | 0.04 | < 0.001 | |

[1] Interaction term between soil temperature and river level

For a deeper understanding of the LMER outputs (Table 1), the individual relationships between surface $CO_2$ fluxes and the
different predictors (soil temperature, river level, and soil moisture) as well as the effect of the interaction between the soil
temperature and river level are visualized in Figure 4. The inclusive r² ($IR^2$) of each predictor is also presented, offering a
measure of the proportion of variance explained by each predictor, including both its direct effects and interactions with other

predictors (Stoffel et al., 2021). In this context, the soil temperature ($IR^2 = 0.225$), soil moisture ($IR^2 = 0.126$), and the quadratic component of the river level ($IR^2 = 0.097$) appear as the primary factors explaining the variance of surface $CO_2$ fluxes, whereas the interaction between soil temperature and river level ($IR^2 < 0.001$), along with the linear component of the river level ($IR^2 = 0.001$), make no meaningful contribution (Figure 4)

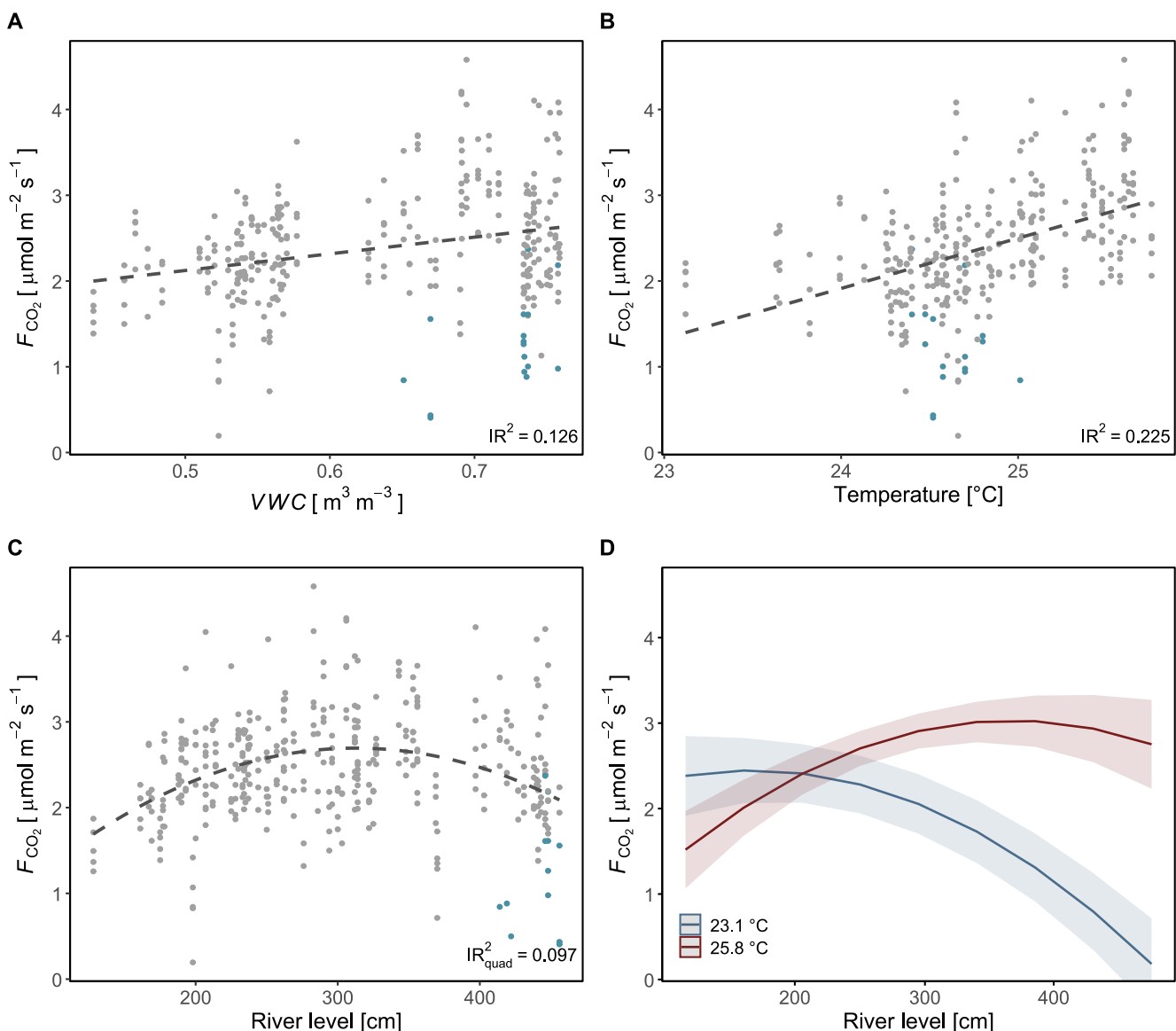

**Figure 4. Individual relationships between soil CO₂ fluxes and the environmental parameters (soil moisture (VWC) (A), soil temperature (B), and river level (C)). Measures taken while the soil chamber was partially flooded are represented in blue. Regression lines are displayed as dashed black lines. The interaction between soil temperature and river level is illustrated. Values were predicted based on the LMER model (Table 1) (D). Inclusive r² (A, B, C) were estimated based on the LMER model (Table 1; Stoffel et al., 2021).**

### 3.2.2. Controls on surface $CO_2$ fluxes at the PFF site

At the PFF site, surface $CO_2$ fluxes did not exhibit statistically significant relationships with pH, river level, carbon to nitrogen ratio (C:N), dissolved organic carbon, or biodegradable dissolved organic carbon. Trends were observed, such as an increase in $CO_2$ fluxes with rising river levels and opposing trends with pH, $CO_2$ fluxes appeared to decrease as pH increased (Supp. Fig. 7). However, these are just visual tendencies and not statistically significant findings.

### 3.3 $\delta^{13}C$ of leaves, litter, soils, soil $CO_2$ flux, and riverine dissolved $CO_2$

The measured $\delta^{13}C$ values increased from leaves over litter and SOC to soil $CO_2$ fluxes and became more positive along this cascade of organic matter transformation (p-values $< 0.05$; Figure 5). $\delta^{13}C$ of leaves ranged from -37.1 to -28.9‰ with a mean of $-33.8 \pm 2.1$‰. The $\delta^{13}C$ signature of litter was between -32.6 and -28.7‰ and, on average $-30.5 \pm 1.0$‰. SOC had $\delta^{13}C$ of -30.1 to -22.3‰, while the mean was $-27.4 \pm 1.9$‰. The $\delta^{13}C$ of soil-derived $CO_2$ ($F_{CO2}$) was in the range of SOC values for the SFF site (Figure 5) and very stable throughout the measurement period (Supp. Fig. 4). Here, measured $\delta^{13}C$ values were -30.2 to -26.5‰ with a mean of $-28.5 \pm 0.8$‰. Contrary, the carbon isotopic composition of $CO_2$ fluxes from the SFF site during flooding was strongly $^{13}C$ enriched with -24.8 to -13.3‰ and an average of $-20.4 \pm 3.4$‰ (p-values $< 0.01$). The $\delta^{13}C$ of the inundated soil $CO_2$ fluxes was higher throughout the whole measurement period (Supp. Fig. 4). The $\delta^{13}C$ value of dissolved $CO_2$ from the adjacent Ruki river was highly stable throughout the measurement period from 2022-10 to 2023-06 ranging from -24.9 to -23.3‰ with a mean of $-24.3 \pm 0.5$‰ (Supp. Fig. 4).

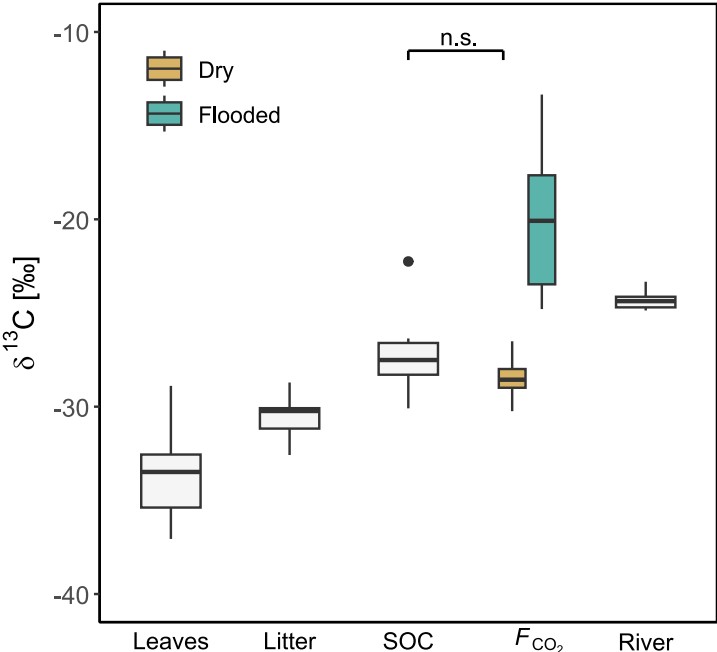

**Figure 5. δ¹³C values of leaves, litter, soil organic carbon (SOC), soil CO₂ flux (F) at the SFF site as well as riverine dissolved CO₂ (Ruki river). Surface CO₂ flux (F) is further separated into dry and inundated based on chamber type (floating, static). Non-significant difference between SOC and dry F$_{co2}$ is indicated by n.s.**

## 4 Discussion

### 4.1 CO₂ fluxes

The surface CO₂ flux dataset from the SFF site, measured for three consecutive years, showed intra-seasonal and interannual variability. However, no clear seasonal patterns were observed (Figure 3E). Baumgartner et al. (2020) showed a similar low seasonality in lowland forests of the Congo Basin, attributing it to the limited rainfall variation between dry and wet seasons. The unclear seasonal pattern of the CO₂ fluxes at the SFF site could be attributed to a lasting effect of the flooding on soil moisture (Docherty and Thomas, 2021) and/or consistent rain events during the whole year (Figure 3A-B). These factors, along with the brief duration of both dry seasons, may lead to soil moisture contents remaining near optimal conditions for vegetation and soil microbes to thrive. Such uniform environmental conditions may maintain autotrophic and heterotrophic respiration at a steady level despite undergoing a discernible dry and wet season cycle. The reported mean flux of $2.36 \pm 0.51$ $\mu mol \ m^{-2} \ s^{-1}$ from the SFF site was lower compared to previous studies in the Congo Basin. These studies found mean values of respectively $3.13 \pm 1.22 \ \mu mol \ m^{-2} \ s^{-1}$, $3.45 \pm 1.14 \ \mu mol \ m^{-2} \ s^{-1}$ in montane and lowland forests (Baumgartner et al. 2020) and $4.07 \pm 0.90 \ \mu mol \ m^{-2} \ s^{-1}$ in a lowland secondary forest of Cameroon bordering the Congo Basin (Verchot et al., 2020).

Compared to similar tropical forest studies, our values are at the low end of the range reported across the pantropical forest realm (Table 2).

The perennially flooded forest site (PFF), located at the interface between terrestrial (forest) and aquatic (stream) ecosystems showed relatively high emissions (4.38 ± 0.64 µmol m$^{-2}$ s$^{-1}$) when compared to other tropical flooded forests (Scofield et al. 2016; Table 2) or those streams draining catchments dominated by seasonally or continually inundated swamp forests (Mann et al., 2014; Alin et al. 2011; Table 2). The elevated $CO_2$ fluxes at the PFF site resulted in higher fluxes relative to the SFF site. Further research is needed to determine whether a greater water depth-integrated respiration (Amaral et al., 2020), a

positive correlation with a larger inundated area (Amaral et al., 2020), prolonged river interactions or other factors explain such difference. In contrast, the SFF site presented reduced $CO_2$ fluxes during the onset of flooding, speculatively due to the inhibitory effect of excessive soil moisture on soil respiration (Courtois et al., 2018; Nissan et al., 2023). A non-significant positive trend between water level and the aquatic $CO_2$ fluxes was visually discernible (Supp. Fig. 7) which is in line with a positive relationship between $pCO_2$ and discharge measured on the adjacent Ruki (Drake et al., 2023). As a constant gas transfer

velocity was used in the present study, short-term changes in aquatic $CO_2$ fluxes reflect the variations in carbon dioxide concentrations ($pCO_2$) in the water. Moreover, the generally low gas transfer velocity (3.5 cm h$^{-1}$) reflects further the very high $pCO_2$ concentrations (10197 – 17260 ppm) measured at the PFF site. These values are significantly higher than the range (3069 – 9088 ppm) found by Drake et al., (2023). However, the adjacent Ruki water has a long transit time compared to a swamp and a stronger current which in turn results in higher $CO_2$ outgassing. Generally, the $pCO_2$ concentration itself is driven

by factors such as terrestrial inputs, gas exchange with the atmosphere, water temperature (gas solubility), water chemistry (pH, alkalinity), and in-stream metabolism (Battin et al., 2023; Hotchkiss et al., 2015; Rocher-Ros et al., 2019).

Finally, the *in-situ* derived gas transfer velocity ($k_x$) expressed as normalized $k_{600}$ (2.95 cm h$^{-1}$) was higher than the global normalized estimate ($k_{600}$) for tropical wetlands (2.4 cm h$^{-1}$; Aufdenkampe et al. 2011). The gas transfer velocity itself changes

by factors influencing the near-surface water turbulence (wind speed, water current velocity). Generally, assuming a constant gas transfer velocity ($k_x$), as applied in this study, has its limitations since it likely varies throughout the year with increased values during the dry season when the water is flowing in the stream bed channel (Alin et al., 2011).

**Table 2. Reproduced mean values of surface $CO_2$ fluxes across various tropical forested environments**

| Country/Basin | Environment | Temporal coverage | $F_{CO_2}$ ($\mu$mol m$^{-2}$ s$^{-1}$) | Source |
|---|---|---|---|---|
| **DRC / Congo Basin** | **Seasonally flooded forest** <br> **Perennially flooded forest** | **3 years** <br> **1 year** | **2.36 ± 0.51** <br> **4.38 ± 0.64** | **This study** |
| DRC / Congo Basin | Montane and lowland (*terra firma*[1]) forests | 3 years, at varying temporal resolution | 3.13 to 3.45 | Baumgartner et al. (2020) |
| DRC / Congo Basin | Lowland (*terra firma*[1]) forest | 16 months, sub-daily resolution | 4.04 ± 1.16 | Daelman et al. (2024) |
| ROC / Congo Basin | Streams (< 100 m wide) draining swamp forests | 3 punctual campaigns over the hydrological year | 3.61 ± 1.46 | Mann, Spencer, et al. (2014) |
| Cameroon | Lowland (*terra firma*[1]) forest | 17 months | 4.07 ± 0.90 | Verchot et al. (2020) |
| Kenya | Montane (*terra firma*[1]) forests | 2-3 months, dry season and transition period | 1.04 to 1.66 | Arias-Navarro et al. (2017); Werner, Kiese, and Butterbach-Bahl (2007) |
| Panama | Lowland poorly drained forest | 3 years | 4.26 ± 0.16 | Rubio and Detto (2017) |
| Brazil / Amazonian Basin | Seasonally flooded forest | From 1 to 2 years, at varying temporal resolution | 2.2[2] to 5.28 | Amaral et al. (2020); Borges Pinto et al. (2018); Zanchi et al.( 2011) |
| Brazil / Rio Negro Basin | Perennially flooded forest | Punctual campaigns (low and high-water periods) | 0.52 ± 0.21 | Scofield et al. (2016) |
| Brazil / Amazonian Basin | Streams (< 100m wide) draining Amazonian wetlands | Punctual field campaigns integrating low and high flow periods | 5.45 ± 3.39 to 5.49 ± 3.16 | Alin et al. (2011); Rasera et al. (2008) |
| Amazonian Basin | Lowland *(terra firma*[1]*)* forest | Variable | 2.30 to 5.30 | (Davidson, Ishida, and Nepstad (2004); Doff sotta et al. (2004); Sousa Neto et al. (2011); Sotta et al. (2007); Garcia-Montiel et al. (2004); Borges Pinto et al. (2018); Janssens, Têtè Barigah, and Ceulemans (1998); Buchmann et al. (1997); Bréchet et al. (2021); Epron et al. (2013); Courtois et al. (2018); |
| Thailand | Lowland (*terra firma*[1]) forest | Punctual measurements over 2.5 years | 6.57 ± 3.42[3] | Adachi et al. (2009) |

| Malaysia | Lowland (*terra firma*[1]) forest | Punctual measurements over 2 and 4 years | $5.32 \pm 2.85$ to $5.7 \pm 1.9$ | Katayama et al. (2009); Ohashi et al. (2007) |
| Australia | Seasonally flooded forest | 13 months | $1.4 \pm 1.0$ / $2.4 \pm 1.4$ (dry season / wet season) | Goodrick et al. (2016) |

[1] *Terra firma* forests refer here to non-flooding forests
[2] Measurements done only during the inundated period
[3] Mean soil respiration for the wet season

## 4.2 Temperature, soil moisture and water level controls

While the observed $CO_2$ fluxes at the SFF site showed no clear seasonal pattern, soil temperature, soil moisture and the river level as proxy of the water level emerged as significant controls. While the positive effect of temperature and soil moisture on soil $CO_2$ fluxes is well known and used to model soil $CO_2$ fluxes (Nissan et al., 2023), the effect of water level is less well understood. The observed quadratic relationship with the water level suggests an optimal water level beyond which further increases lead to reduced $CO_2$ fluxes. This optimal point speculatively corresponds to the shift to water saturated conditions in the organic-rich surface soil transitioning from oxic to anoxic conditions. A negative effect of the water level beyond a critical threshold aligns well with the results of Goodrick et al., (2016). That study found maximal soil $CO_2$ fluxes associated with a water level between 1.5 and 2 m below ground and minimal fluxes when the water level was within 0.15 m of the surface for a tropical riparian swamp forests in Australia (Goodrick et al., 2016). Similarly, Rubio and Detto (2017) found a quadratic relationship between $CO_2$ fluxes and soil water content in the Amazonian basin. $CO_2$ fluxes can be reduced in both high and low soil water content, and fluctuations in water level introduce additional factors beyond its influence on soil saturation. Both heterotrophic and autotrophic soil respiration are reduced under dry conditions due to limited microbial activity and reduced photosynthetic activity through stomatal closure (Baumgartner et al. 2020). In our study, the lowest flux event recorded in July 2022 coincided with a marked decrease in soil moisture. This suggests that, during this event, the reduced soil moisture levels became a limiting factor for supporting soil respiration. Conversely, increased soil moisture generally enhances respiration. This was generally the case during our study period, as evidenced by the positive correlation between soil moisture and surface $CO_2$ fluxes (p-value $< 0.05$; Table 1). However, excessively high moisture conditions (due to strong rain events or high water level during flooding) can also hinder substrate decomposition by physically impeding the diffusion of atmospheric oxygen and respired $CO_2$ through the soil pores, thereby limiting both the production and diffusion of $CO_2$ (Courtois et al., 2018; Nissan et al., 2023). This could explain the temporary decrease in $CO_2$ fluxes observed at the onset of the flooding period (Figure 3). Furthermore, fluctuations in the water level can influence soil respiration through physical processes like flushing out soil $CO_2$ during rising phases, enhanced lateral movement of dissolved $CO_2$, as well as air ingress and redistribution of organic material during receding phases (Dalmagro et al., 2018; Goodrick et al., 2016). Finally, the positive interaction between

soil temperature and water level (p-value < 0.05; Table 1) suggests that higher temperatures will reinforce the effect of the water level and shift the maximum soil flux towards higher water levels delaying its inhibitive effect (Figure 4).

Nevertheless, it is important to note that both the water level and soil moisture measurements exhibit seasonal patterns but do not capture well the short-term changes of the surface $CO_2$ fluxes at SFF site. Furthermore, the $CO_2$ fluxes exhibit unclear seasonal pattern (Supp. Fig. 5B). This suggests that other factors, such as aboveground inputs from vegetation, river sediment deposition, and rain-induced events, may significantly influence surface $CO_2$ fluxes, both in the short term and at seasonal timescales. Additionally, it is important to stress that using river level as a proxy for water level at the SFF site presents limitations such as neglecting local topography or soil characteristics. Thus, fortnightly variations in soil $CO_2$ fluxes may not be fully captured by this proxy, as local hydrological dynamics might differ from those of the broader river system. Hence, this method may not fully capture the dynamics of the water level and its influence on surface $CO_2$ fluxes.

Overall, while soil moisture content and temperature are often considered primary drivers of soil $CO_2$ fluxes (Courtois et al., 2018; Nissan et al., 2023; Oertel et al., 2016), our findings also indicate that incorporating water level can help to unravel the variability of the fluxes for lowland forests with shallow water tables.

At the PFF site, on the other hand, we did not find any statistically significant relationships between potential drivers (DOC, BDOC, river level, pH, C:N) and $pCO_{2w}$. This suggests that the chemical composition of the water is relatively homogenous throughout the year and that allochthonous rather than autochthonous processes determine $pCO_{2w}$ concentrations.

### 4.3 Isotopic indicators

The general carbon isotopic composition of plant tissue is determined by the degree of [13]C discrimination at the leaf level (Brüggemann et al., 2011). Due to the high photosynthetic activity of tropical plants, [13]C discrimination is also high, resulting in very negative $\delta^{13}C$ values at the leaf level as observed in this study -37.06 to -28.89‰). As the C moves across the various ecosystems C pools, the substrate becomes gradually enriched in [13]C due to kinetic isotope fractionation. In the case of the studied SFF site, a total [13]C enrichment of 5.27‰ was observed when moving down the cascade from leaves, litter to SOC and respired $CO_2$ under dry conditions (p-values < 0.05; Figure 5). Particularly interesting here is the absence of [13]C fractionation between SOC and soil respired $CO_2$, which might initially be interpreted as a result of closed system dynamics where the substrate is limited, and organic decomposition tends to be complete. However, soil respired $CO_2$ is a two-component flux, comprised of heterotrophic and autotropic respiration. In other words, SOC is not the sole factor governing soil respired $CO_2$. Indeed, autotropic respiration is to a large degree fuelled by recently photosynthesized [13]C depleted carbon (Högberg et al, 2001, Barthel et al., 2011) which in turn can decrease the overall soil respired $\delta^{13}C$ value relative to SOC (depending on the relative contribution of autotrophic vs heterotrophic soil respiration). Transport rates from above to belowground can reach up to 0.5 m h$^{-1}$ (Kuzyakov and Gavrichkova, 2010). Thus, whether the similar $\delta^{13}C$ values between SOC and respired $CO_2$ are driven by substrate limitation or a strong influence of autotrophic respiration requires further investigation.

The highest $^{13}C$ enrichment observed was from $CO_2$ emitted during flooding at the SFF site (-20.4‰; p-values < 0.05). These δ$^{13}C$ values were even higher than the δ$^{13}C$ values measured in the adjacent Ruki river (-24.30 ‰; p-value < 0.05). The reason for such highly $^{13}C$ enriched $CO_2$ outgassing during inundation remains unclear but given that the water in the inundated forest likely experiences relatively long residence times compared to the river, the outgassed $CO_2$ might become this heavily $^{13}C$-enriched due to extensive outgassing. Moreover, the standing water allows the growth of methanogenic archaea which use simple carbon compounds such as acetate as electron donors (Conrad et al., 2021). The $CO_2$ molecules obtained from acetate cleavage is another fractionation process which potentially influences the overall isotopic composition of outgassed $CO_2$. Lastly, as the inundation of the SFF site is mainly driven by backflow from the river system, the dissolved $CO_2$ in the inundated water could be a mix of riverine and locally soil-respired $CO_2$ that undergoes further *in-situ* $^{13}C$ enrichment.

**5 Conclusion**

This study presents a multi-year dataset of $CO_2$ fluxes from two forested wetland sites along a flooding gradient : a seasonally flooded forest (SFF) and a perennially flooded forest (PFF). While exhibiting short-term and interannual fluctuations, $CO_2$ fluxes showed limited seasonal patterns. At the SFF site, surface emissions increased with rising soil moisture and temperature, while the water level demonstrated a significant quadratic relationship. Despite the significant sensitivity to environmental conditions over the observation period, the short-term variability observed at both sites, as well as the interannual variability at the SFF site, were incompletely explained, suggesting the influence of additional factors in regulating emissions.

Our results emphasize that water level, alongside soil temperature and soil moisture, significantly affects surface $CO_2$ fluxes in lowland areas with shallow, fluctuating water tables. Future research should include direct measurements of the water level over the entire hydrological year to elucidate the temporal dynamics of this relationship. Overall, the reported measurements contribute to filling the data gap for soil respiration rates of tropical forests in the Congo Basin and provide baseline fluxes for parametrizing earth system models.

**Data Availability**

The datasets generated in this study have been deposited in the Zenodo repository (https://doi.org/10.5281/zenodo.15051088) and are available from the corresponding author upon request. Additionally, the data used in this study has been made available through the soil respiration database (SRDB; Jian, J. et al. 2021).

## Author contributions

Mbarthel, MBauters, TWD, KVO, and JS were responsible for study design and conceived the study. Fieldwork was conducted by Mbarthel, Mbauters, TWD, SB, NBM, JC, ADC, and CE. Lab work was conducted by Mbarthel, RAW, SB and JC. Data analyzes and interpretation was performed by ACHJ, ADC, Mbauters and Mbarthel. ADC and ACHJ wrote the manuscript with contributions from all co-authors.

## Competing Interests

At least one of the (co-)authors is a member of the editorial board of Biogeosciences.

## Acknowledgements

The authors would like to thank the Trans-African Hydro-Meteorological Monitoring Observatory (TAHMO) for providing meteorological data from the investigated site and the ICCN park guards for assistance in the field. The authors would like to further thank Annika Ackermann from the Grassland Science group (Prof. Buchmann) at ETH Zurich for the stable isotope analysis of plant and soil materials.

## Funding Statement

The core funding of ETH Zurich financed this study. MBarthel received funding from the Swiss National Science Foundation (IZSEZ0_186376 / 1).

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
