# Peer review of "Aquatic and Soil CO2 Emissions from forested wetlands of Congo's Cuvette Centrale"

_EGUsphere, 2024_

## Author Response (AR1)

**Anonymous reviewer 1:**

The manuscript presents valuable data on forested wetlands in the Congo, addressing a significant gap in current knowledge and demonstrating how different flooding regimes impact surface $CO_2$ fluxes. I particularly appreciate the detailed descriptions in the Methods section. However, I believe the Results section would benefit from a more thorough discussion, and the Discussion section could be further strengthened. Additionally, there are several formatting issues, such as mismatches between in-text figure numbers and the actual figures, as well as inconsistent use of subscripts. Below are my detailed comments.

Dear Reviewer, thank you for your time and positive, valuable as well as detailed feedback on our manuscript. Please find below how each of your comments has been integrated into the new version of the manuscript.

Line 30: I'd suggest using the full term for $pCO_2$ before using the acronym.

The sentence has been changed as follows: due to very high partial pressure of $CO_2$ ($pCO_2$) values measured in the flooding waters.

Line 55-57: The authors mention that 'probably only one study' has looked into GHG emissions from Congo's forested wetlands, specifically methane (Tathy et al. 1992). However, it seems there is another study that also investigates GHG emissions, including both methane ($CH_4$) and nitrous oxide ($N_2O$), from forested wetland soils in the Congo Basin (Barthel et al., 2022, Nature Communications).

Thank you for pointing this out. The sentence has been modified:

'Despite its immense global importance, only two studies, to the best of our knowledge, have been looking into GHG emissions from Congo's wetlands ).

Line 71-73: I suggest moving the description of measurement variables to the methods section. This would help streamline the Introduction and ensure methodological details are presented in the appropriate context.

The description of the measurement variables has been removed from that paragraph and explained in detail in the methods section.

Line 94: The sentence suggests that the forest is flooded because of its proximity to the river, which may not fully capture the cause of the seasonal flooding. It would be more accurate to say that the forest is flooded due to increased river flow during the rainy season or similar hydrological events. I recommend rephrasing the sentence to reflect the influence of the rainy season on the river's water levels, which causes the flooding.

Indeed, the sentence does not consider the different possible causes of the seasonal flooding. As the different origins of the flooding water (rainfed or riverfed) are already discussed in section 2.6 Water level, we suggest simplifying the sentence as follows :

'The SFF site is seasonally flooded from about December to January (~ 2 months).'

Line 99: It seems that the sentence doesn't correspond to the content of Figure 2.

Thank you for pointing this out. The reference has been corrected to Figure 3. Our apologies for the issues with the referencing of the figures. There was a last-minute change that led to this mix-up of figure assignments.

Line 116-118: I'd suggest clarifying whether the reference to Drake et al. (2023) is meant to support the statement about the previous analyses showing TDN consistently comprising 90% of DON, or if it only pertains to the specific methods and calculations. It might be helpful to provide a reference for the previous analyses as well to avoid confusion.

We modified the sentence as follows to overcome confusion:

'Previous analyses (Drake et al. 2023) showed that TDN consistently comprised an average of 90% of DON and thus reflected well the relative changes of DON concentrations. The specific methods used for sample processing and analysis as well as the calculations are described in Drake et al., (2023).'

Line 126: Please clarify the abbreviations "h" and "ø" for better readability. It would be helpful to define these terms for readers who may not be familiar with them.

Modified as follows:

'A total of six polyvinylchloride soil flux chambers (height = 0.3m, diameter = 0.3m) were installed in November 2019'

Line 135: Please use the full term for "h" here.

Modified as follows: 'were taken at timesteps of 20 min throughout 1 hour.'

Line 144-145: Could you please clarify why data with such a low $r^2$ threshold were retained, despite the strong regression fit observed in most of the data?

Most fluxes with a low $r^2$ are related to low flux rates and not because of methodological or technical issues. Eliminating cases with low $r^2$ would bias the data to high fluxes. Therefore, we retained also fluxes with low $r^2$ values.

The following sentence was added to clarify this for the readers :

'Such a low $r^2$ threshold was maintained because fluxes with low $r^2$ values are typically related to low flux rates rather than due to methodological or technical issues. Increasing the threshold would introduce a bias toward higher fluxes in the data.'

Line 166: Remove the space before the colon.

Corrected.

Line 172: Please add a space between "50" and "μL" for consistency with units.

Corrected.

Line 179: The abbreviation "GC" should be defined earlier as "gas chromatograph" in line 177.

Corrected as follows: 'Gas samples were analyzed at ETH Zurich using a gas chromatograph (GC, Bruker, 456-GC, Scion Instruments, Livingston, UK)'

Line 184: Remove the space after the dash.

Corrected

Line 189: Please use the full term "V-PDB" when first mentioned.

The sentence was corrected as follows: 'Post-run off-line calculation and drift correction for assigning the final $\delta^{13}C$ values on the Vienna Peedee Belemnite (V-PDB) scale was done following the 'IT principle' (R. A. Werner and Brand 2001).

Line 206-212: Water table already implies the level of water beneath the ground so "level" is redundant. Since you seem to be referring to the river, "water level" may be a more appropriate term.

This paragraph was revised: 1) The redundancy of the term "level" when referring to the water table was removed. In accordance with the feedback of Reviewer #2, to avoid confusion, the term "water level" rather than "water table" will be consistently used in the manuscript 2) The level of the river is referred to as "river level".

Line 208: Could you clarify the distance between the Congo River and the study site, or explain how it is hydrologically connected to the Ruki River?

The Ruki river at the point of sampling is directly hydrologically connected to the Congo River with a distance of approximately 4000m from the Ruki River gauge to the confluence with the Congo River. This is exemplified by the direct relationship between Congo River and Ruki River water levels (Supplementary Fig. 3). The distance from the SFF study site itself to the Ruki River is 755m.

The following sentence was modified to clarify this :

'This data was extracted from an almost continuous record of water gauge readings, collected in vicinity of the SFF site (~ 4 km) by the Congolese public institution, *Régie des Voies Fluviales*, since 1913.'

Line 210: DRC is not defined earlier in the text.

Corrected as follows: '...water levels of riverine locations in the Democratic Republic of the Congo (DRC) correlate...'

Line 215: I'd suggest changing the heading from "Statistics" to "Statistical Analyses" for better clarity and accuracy.

Changed as suggested.

Line 218: "Added" might be a better word than "tested" as the quadratic term was included in the model to account for the non-linear effect.

Corrected as follows: 'Hence, a quadratic term was added to account for the non-linear effect.'

Line 232: Remove the space after 'tydr v1.3.0' and before the comma.

Space removed.

Line 236-248: For this section, I'd suggest first describing the general climate characteristics at the study site, such as the timing of the dry and wet seasons, to better contextualize the variables. Additionally, I recommend incorporating the water level into this section, potentially alongside Figure 3. Although you mentioned that in-situ measurements were not available and the Congo River water level was used as a proxy, you also noted in Section 2.6 that riverine water levels are more influenced by river hydrological dynamics than rainfall. Therefore, you might consider using a more general heading, like "Environmental Conditions". For line 238, should it refer to Figure 3A instead of 2A? Lastly, in lines 247-248, please combine the sentences for better continuity instead of starting a new line.

The references were corrected to Figure 3. The sentences were combined in lines 247-248. The section heading has been changed to "Environmental Conditions".

The first sentences (line 236) were modified to contextualize the information. We focus on the flooding period (Dec-Jan) rather than the wet/dry season. Frequent rain events (see Figure 3) render the region as relatively wet throughout the year. We mention now in the text which months are considered to be the dry and wet seasons as follows:

'The long dry season in Mbandaka is considered from July to August whereas the short dry season spans between January and February. However, frequent rainfall as shown in Figure 3 renders the region as relatively wet throughout the entire year. Annual precipitation was the highest in 2020 with 1855 mm and lowest in 2022 with 1417 mm (self-measured; Figure 2). The flooding period at the study site is typically centered around December and January. '

Finally, Figure 3E (numbering in the updated version, previously 3C) was updated to include the river level as suggested. After further modifications of the figure to integrate the feedback from Reviewer 2, the caption now reads:

**Figure 1. Weekly precipitation, volumetric soil water content, temperature, and $CO_2$ fluxes. (A) The sum of the weekly precipitation [mm] (blue) obtained from the Trans-African Hydro-Meteorological Monitoring Observatory and mean volumetric soil water content [$m^3$ $m^{-3}$] measured with soil moisture sensors (ECH$_2$O 5TM = solid line, TMS-4 dataloggers = dotted line). (B) Distribution of volumetric soil water content [$m^3$ $m^{-3}$], both sensor types combined. (C) Mean weekly air temperature [°C] (gold) was obtained from the Trans-African Hydro-Meteorological Monitoring Observatory. The mean weekly soil temperature [°C] was measured with soil temperature sensors (ECH$_2$O 5TM = grey solid line, TMS-4 dataloggers = grey dotted line). (D) Distribution of air and soil temperatures [°C], both sensor types combined. (E) Measured surface $CO_2$ fluxes (cross) [$\mu mol$ $m^{-2}$ $s^{-1}$] from the SSF site (brown) and calculated $CO_2$ fluxes from the PFF site with a K of 3.5 cm $h^{-1}$(pink). Calculated weekly means (line) and the standard error of the mean are displayed. Blue shading represents river levels (see Section 2.6), while grey bands indicate flooding periods (Dec-Jan) at the SFF site. The displayed time series are discontinuous due to fieldwork constraints (see Section 2.2.) (F) Distribution of surface $CO_2$ fluxes at the PFF and SFF sites.**

And here is the updated figure 3 :

[Figure]

Line 249-256: This section presents CO₂ flux data from two sites (PFF and SFF), showing variations over time. However, it doesn't explicitly explain the drivers behind these fluctuations. It would be useful to explore how these CO₂ flux variations correlate with specific environmental variables (e.g., temperature, precipitation, soil moisture, water level) or if they follow consistent seasonal patterns.

The text was modified to describe more explicitly the variation of the CO₂ flux and acknowledge the significant correlations with environmental variables. For readability, we suggest exploring further the correlation between drivers and flux fluctuations in the following results section 3.2.1. This section has also been revised to not only describe the signs and relative importance of the main environmental drivers but also to examine the relative contributions of each variable in explaining flux variations.

The start of the section (LN 249) now reads:

'Over the observation period, $CO_2$ fluxes from the PFF site were higher than from the SFF site (Figure 2E). At both sites, $CO_2$ fluxes exhibited intra-annual variability. However, distinct seasonal patterns were not clear. Notably, at the SFF site, the onset of flooding appeared to induce a decline in fluxes. Furthermore, among the environmental variables, $CO_2$ fluxes exhibited significant correlations with soil moisture, soil temperature, and river level ('Table 1).'

And the section 3.2.1. has been completed with:

'For a more complete understanding of the LMER model ('Table 1), the individual relationships between surface fluxes and the different predictors (soil temperature, river level, and soil moisture) as well as the effect of the interaction between the soil temperature and river level are visualized in Figure 3. The inclusive $R^2$ ($IR^2$) of each predictor is also presented, offering a measure of the proportion of variance explained by each predictor, including both its direct effects and interactions with other predictors (Stoffel et al., 2021). In this context, soil temperature ($IR^2 = 0.225$) soil moisture ($IR^2 = 0.126$), and the quadratic component of the river level ($IR^2 = 0.097$) appear as the primary factors explaining the variance of surface $CO_2$ fluxes, whereas the interaction between soil temperature and river level ($IR^2 < 0.001$), along with the linear component of the river level ($IR^2 = 0.001$), make no meaningful contribution (Figure 3).'

To account for the estimation of the inclusive $R^2$, the statistical methods were also completed as follows :

'Marginal and conditional $R^2$ values for mixed effects were calculated using Nakagawa et al., (2017), inclusive $R^2$ estimated with *partR2* package (Stoffel et al., 2021) and p-values using Satterthwaite's approximation with the *lmerTest* package (Kuznetsova et al., 2017).'

Line 250, 254 and 256: Same comment as Line 238—should it be Figure 3C instead of 2C?

Thank you for pointing this out, the references were corrected to Figure 3.

Line 260: Figure 3 presents important data, but the current color scheme and legend structure could be improved for better clarity:

- Subplot A: The colours for the three lines (precipitation and two soil moisture sensors) are too similar, making it difficult to distinguish between them.

- Subplot B: The two lines representing soil temperature from different sensors (ECH2O 5TM and TMS-4 dataloggers) also have very similar colours, which may confuse readers.

- To enhance readability, it would be beneficial to include legends for each subplot.

As shown earlier, Figure 3 has been modified: colours and style are now more distinct and harmonized between subplots, and legends are included for each subplot. In response to the comment on Lines 236-248, the water level was also integrated into the figure.

Line 279: Are the fixed effect estimates presented in Table 1 standardized or are they unstandardized? Clarifying this would help in interpreting the relative importance of the predictors and their effects on soil $CO_2$ fluxes.

The estimates presented are standardized. The legend of the table was modified as follows:

'**Table 1. Fixed effect estimates for surface $CO_2$ fluxes at SFF site including river level, soil temperature, and soil moisture as standardized predictors, allowing comparison of their relative importance. For each effect, standard error**

and p-values (Sattherhwaite's method) are estimated, as well as the marginal (m) and conditional (c) $R^2_{adfj}$ (Nakagawa, Johnson, and Schielzeth 2017).

Line 287-290: The text appears to be the caption for Figure 4, please remove it from the text.

The legend text was removed from the main text.

Line 294-295: There appears to be a mismatch between the subplot labels (A and B) in the Figure 4 caption and the content. Also, please use the correct subscript for "$CO_2$".

Mismatched labels and subscripts were corrected.

Line 309: Figure 3 doesn't appear to show δ13C values.

The reference was corrected to Figure 5.

Line 325-334: The authors compare $CO_2$ fluxes from both seasonal and perennial flooded forest sites with those from other reported sites. However, the reasons for the higher emissions at the PFF compared to the SFF are unclear. It would be beneficial to discuss this, as it could provide further insight into the differences between the two forest types.

This is a good point raised by the Reviewer. In order to discuss it, this section now reads as follows:

'The perennially flooded forest site (PFF), located at the interface between terrestrial (forest) and aquatic (stream) ecosystems showed relatively high emissions ($4.38 \pm 0.64$ µmol m$^{-2}$ s$^{-1}$) when compared to other tropical flooded forests (Scofield et al. 2016; **Error! Reference source not found.**) or those streams draining catchments dominated by seasonally or continually inundated swamp forests (Mann et al., 2014; Alin et al. 2011; **Error! Reference source not found.**). The elevated $CO_2$ fluxes at the PFF site resulted in higher fluxes relative to the SFF site. Further research is needed to determine whether a greater water depth-integrated respiration (Amaral et al. 2020), a positive correlation with a larger inundated area (Amaral et al. 2020), prolonged river interactions or other factors explain such difference. In contrast, the SFF site presented reduced $CO_2$ fluxes during the onset of flooding, speculatively due to the inhibitory effect of excessive soil moisture on soil respiration (Courtois et al. 2018; Nissan et al. 2023). '

Line 340-341: It would be helpful to provide references for these factors influencing $p$CO2 concentrations.

References were added as follows:

'Generally, the $p$CO$_2$ concentration itself is driven by factors such as terrestrial inputs, gas exchange with the atmosphere, water temperature (gas solubility), water chemistry (pH, alkalinity), and in-stream metabolism (Rocher-Ros et al. 2019; Hotchkiss et al. 2015; Battin et al. 2023).'

Line 346: I did not see the plot of water level and $p$CO2 in Figure 5. Could you please clarify if this plot is included or if it is shown in a different figure?

We apologize for this confusion. The statement refers to the plot depicting $CO_2$ fluxes and water levels (Figure 5A), and by extension, to $p$CO$_2$ values. The use of a constant gas transfer velocity to calculate Fco$_2$ in the study implies that variations in $CO_2$ fluxes correspond to variations in pCO$_2$.

The sentence was revised for clarity and moved to line 337 to improve readability and text logic:

'A non-significant positive trend between water level and the aquatic $CO_2$ fluxes was visually discernible which is in line with a positive relationship between $p$CO$_2$ and discharge measured on

the adjacent Ruki (Drake et al., 2023). As a constant gas transfer velocity was used in the present study, short-term changes in aquatic $CO_2$ fluxes reflect the variations in carbon dioxide concentrations ($pCO_2$) in the water. Moreover, the generally low gas transfer velocity (3.5 cm h$^{-1}$) reflects further the very high $pCO_2$ concentrations (10197 – 17260 ppm) measured at the PFF site.'

Line 350: Please use a subscript for "CO2" in the table caption.

Corrected.

Line 379: Supp. Fig. 6 does not show seasonal $CO_2$ fluxes.

Corrected to Supp. Fig. 5B

Line 379-380: The authors suggest that factors like aboveground inputs, deposition, and rain-induced events may influence soil $CO_2$ fluxes. While a linear relationship between the Congo and Ruki river levels is noted, and previous research indicates that water levels in the Cuvette Centrale correlate more with river dynamics than rainfall, there are still uncertainties to consider:

- The study site is approximately 1 km from the Ruki River, and local topography and soil characteristics could affect how well river levels represent the actual water table at the forest site.

- Short-term variations in soil $CO_2$ fluxes may not be fully captured by this proxy, as local hydrological dynamics might differ from those of the broader river system.

- The lack of correlation between water table changes and $CO_2$ fluxes may reflect limitations of the proxy measurement rather than a true absence of a relationship.

While you made a reasonable methodological choice using available data, it might be possible to acknowledge that the study site's water level could differ from the river proxy. This limitation should be considered when interpreting the results, especially regarding the influence of water table dynamics on soil $CO_2$ fluxes.

Thank you for pointing out this methodological limitation. The mentioned limitations were more explicitly acknowledged as follows:

'Nevertheless, it is important to note that both the water level and soil moisture measurements exhibit seasonal patterns but do not capture well the short-term changes of the surface $CO_2$ fluxes at SFF site. Furthermore, the $CO_2$ fluxes exhibit unclear seasonal pattern (Supp. Fig. 5B). This suggests that other factors, such as aboveground inputs from vegetation, river sediment deposition, and rain-induced events, may significantly influence surface $CO_2$ fluxes, both in the short term and at seasonal timescales. Additionally, it is important to stress that using river level as a proxy for water level at the SFF site presents limitations such as neglecting local topography or soil characteristics. Thus, fortnightly variations in surface $CO_2$ fluxes may not be fully captured by this proxy, as local hydrological dynamics might differ from those of the broader river system. Hence, this method may not fully capture the dynamics of the water level and its influence on surface $CO_2$ fluxes.'

**Anonymous Reviewer 2:**

Dear Editor, Dear Authors, it has been a pleasure to review the work by Clippele and colleagues on "Aquatic and Soil CO2 Emissions from forested wetlands of Congo's Cuvette Centrale". I

congratulate the authors on collecting a CO2 flux dataset from the Congo Basin spanning three years at the interface between aquatic and terrestrial landscapes in tropical forests. Such rich datasets remain scarce in the tropics. Thus, this manuscript and emerging insights from the presented data shed new insights on the contribution of these inadvertently under-sampled regions in the Congo Basin to the overall tropical forest CO2 budget. The manuscript should be of interest to the readership community of Biogeosciences. The manuscript is well written, and I recommend its consideration for publication after integrating some minor suggestions and clarifications from the authors.

Dear Reviewer, thank you for your valuable and positive comments and for taking the time to give detailed feedback. Please find below our clarifications and changes implemented following your inputs.

Specific comments

LN 23, why do you think there were no discernible seasonal differences in CO2 fluxes across the two sites? For instance, could it be that in seasonally flooded forests, the effect of flooding on soil moisture during the wet season persists through the dry season, keeping soil moisture content near optimal for microbial and plant activity? This probably could dampen the effect of seasonality on the measured CO2 fluxes at both sites. You need to shed more light on the lack of seasonality effect on CO2 fluxes since you already mentioned the positive correlation between measured CO2 fluxes and moisture content at the seasonally flooded sites. Please shed more light on this.

This is a very important point mentioned by the reviewer. The combination of a lasting effect of the flooding, a relatively high-water table, and consistent rain events during the whole year could indeed lead to soil moisture contents remaining near optimal conditions for vegetation and soil microbes to thrive. Such stable environmental conditions may likely maintain autotrophic and heterotrophic respiration at a steady level despite undergoing a discernible dry and wet season cycle. However, we would like to acknowledge that our available dataset does not allow us to draw a definitive conclusion regarding this assumption.

In accordance with the present feedback as well as the one from Reviewer 1, this is explored in greater detail in the discussion of the paper and is now specifically mentioned in section 4.1.:

'The surface $CO_2$ flux dataset from the SFF site, measured for three consecutive years, showed intra-seasonal and interannual variability. However, no clear seasonal patterns were observed (Figure 2C). Baumgartner et al. (2020) showed a similar low seasonality in lowland forests of the Congo Basin, attributing it to the limited rainfall variation between dry and wet seasons. Similarly, the unclear seasonal pattern of the $CO_2$ fluxes at the SFF site could be attributed to the lasting effect of the flooding (Docherty and Thomas 2021) and/or consistent rain events during the whole year (Figure 2A-B). These factors, along with the brief duration of both dry seasons, may lead to soil moisture contents remaining near optimal conditions for vegetation and soil microbes to thrive. Such uniform environmental conditions may maintain autotrophic and heterotrophic respiration at a steady level despite undergoing a discernible dry and wet season cycle. '

LN 25, can you talk of water table depth when the site is flooded? Maybe ponding depth is more appropriate

We agree with the reviewer that the water table is a misleading term as it refers to the groundwater table. However, we also refrain from using ponding depth as we need to cover the range above-belowground.

For readability, we implemented the following approach:

- When referring to the water level at the SFF site (either above or below ground), we used the term 'water level'. We believe that it is a good term to cover the range of water height above and belowground.
- When referring specifically to the dataset of river gauge measurements or the level of the river (i.e. in section 3.2.1.), we used the term 'river level'.

LN 26-28. What do we take away from the progressive enrichment of pools with 13C? How about the implication of no significant differences in the enrichment between 13C in SOC and respired CO2 pools?

The absence of enrichment between SOC and respired $CO_2$ could be due either to a substrate limitation and closed system dynamics or to an important contribution of the autotrophic component of soil respiration (that would decrease the overall soil respired $\delta^{13}C$ value based on the relative contribution of autotrophic and heterotrophic component). However, the set-up of our study did not allow us to discriminate between both options.

This sentence was added to the abstract to further detail this:

'This lack of enrichment can be attributed to either a significant contribution from the autotrophic component of soil respiration or a result of closed system dynamics.'

LN 29-30. What is the extent of tropical wetlands in the Congo Basin? I think it would be valuable if you could somehow estimate the area coverage of these wetlands and what not measuring from these areas would mean for the total ecosystem CO2 flux budget.

Bwangoy et al., (2010) estimated the wetland cover in the Congo Basin at 359,556 $km^2$ (36% of the total watershed) whereas Fatras et al., (2021) estimated a size of 332,620 $km^2$ with a mean flooded area of 89,408 $km^2$ in the Congo Basin. Furthermore, the Congo has been found to hold the largest peatland complex on Earth (167,600 $km^2$; Crezee et al. 2022), exemplifying the importance of the region for $CO_2$ emissions and the global carbon budget.

The second paragraph of the introduction now explicitly refers to the wetlands cover and their importance for global $CO_2$ emissions. It now reads as follows:

'Wetland cover in the tropical Congo Basin is estimated to range between 332,620 and 359,556 $km^2$ (Bwangoy et al. 2010; Fatras et al. 2021). This area includes the *Cuvette Centrale*, which spans approximately 167,600 $km^2$ and hosts lowland and swamp forests, including the largest peatland complex across the tropics (Crezee et al. 2022). With catchment drainage from north and south of the equator as well as sustained rainfall at the center of the basin (Runge 2007; Breitengroß 1972), the *Cuvette Centrale* shows near permanent inundation. Characterizing $CO_2$ fluxes in this extensive region is especially important since inland waters are increasingly recognized as significant sources of greenhouse gases (GHG) within the terrestrial landscape (Bastviken et al. 2011; Drake, Raymond, and Spencer 2018; Borges, Darchambeau, et al. 2015; Rosentreter et al. 2021) and notably in global carbon dioxide emissions (Raymond et al. 2013).'

LN 40. "up to 64% of global soil respiration" sounds better.

The sentence has been modified as follows: Indeed, tropical regions are estimated to contribute up to 64% of global soil respiration,

LN 49. Please provide the exact amount of C emitted by Congo Basin inland waters compared to the Amazon instead of just saying "more C."

The authors of the referenced study (Alsdorf et al., 2016) based their assertion on a comparison between carbon emission estimates for the Amazon Basin (Melack, 2016) and those for the Congo Basin (Borges, 2015). They argue that while the estimates are similar (~480 Tg C/yr), the difference in basin size (6 vs. 3.6 million km$^2$) suggests that inland waters in the Congo Basin may emit more carbon per unit area than those in the Amazon Basin. However, the authors acknowledge that these comparisons are based on extrapolations from limited datasets, particularly for the Congo Basin, and should be considered as an indication of the potential importance of the Congo Basin regarding carbon dynamics rather than final estimates. Hence, we respectfully suggest refraining from including exact figures as these remain very speculative.

LN 91-92, I would also add information on base saturation to the brackets since it is the premise on which soil is classified as either eutric or dystric. Could you also reference the soil classification system you used, whether it is the WRB 2014 or 2022, etc.? I am also curious to know why there is no mention of stagnic properties in the prefix or suffix qualifiers of the main soil reference group, yet the study area is seasonally flooded. What is the depth of the water table? Please indicate it as well.

Unfortunately, no water table measurements were available for the whole measurement period. However, a subsequent study at the same site measured a range of -74.5 to 113 cm from early November to March. Regarding the classification of the soil, we have no measures of base saturation, it is based on the EU Soil Atlas of Africa. Additionally, the classification is based on WRB 2014 and the qualifiers are not provided.

LN 95-96, I am also curious to know why the sensors that logged soil-environmental parameters were installed at a depth of 30 cm instead of close to the surface where you measure the soil-atmospheric CO2 flux.

Thank you for pointing this out. When setting up the field trial we installed a sensor profile at 5, 10, and 30cm from the surface to have a detailed measurement from the biologically most active layer. Unfortunately, this sensor setup was malfunctioning soon after installation due to the high humidity and termite activity. As the 5 cm sensor had missing data, the data from the other depth layers was used. These issues were resolved after using the TOMST sensors which integrate soil moisture measurements over 14cm topsoil and measure soil temperature at 8 cm depth.

We amended the text as follows:

'At the SFF site, combined soil moisture and temperature sensors (ECH$_2$O 5TM, Meter Group, Inc. USA) connected to loggers (Em50, Meter Group, Inc., USA) were installed at 10 and 30 cm depth, respectively. The data was recorded every 6 h. Unfortunately, one logger was stolen and the other logger stopped working during deployment; thus, data is only available from November 2019 to July 2020 (Figure 2). Afterward, TMS-4 dataloggers (TOMST, Czechia) were installed in December 2020 to record surface volumetric soil water content (0-14cm) and soil temperature at 8 cm depth in 15-minute intervals.'

LN 126 Could you also mention the depth of installation of the chamber bases/collars? Were the chambers left in place for the entire measurement period?

The chamber bases were installed about 5 cm deep and left in place for the entire measurement period with the chambers remaining open except during sampling. This information has been added to the method section. It reads now:

'A total of six polyvinylchloride soil flux chambers (height = 0.3m, diameter = 0.3m) were installed in November 2019 at the SFF site. The chamber bases were inserted approximately 5 cm into the

ground and remained in place throughout the measurement period, with the chambers left open except during sampling.'

LN 135: Longer chamber closure times are not recommended. Pavelka et al. (2018) (doi: 10.1515/intag2017-0045) recommended a maximum chamber closure time of 45 minutes. Could you add a few details in the methods explaining why you chose the maximum closure time of 60 minutes?

Thank you for pointing us to this nice publication. We decided to have such a long chamber closure time b/c we were also interested in measuring the isotopic composition of $CO_2$ released from the surface. Thus, to reach a good signal-to-noise ratio on our IRMS with this chamber size we opted for a longer closure time. We are aware that longer closure times lead to unwanted saturation effects but the near-perfect linear regressions of the concentration vs. time correlations gave us confidence that this was of minor importance. We added the following line to the methods:

"A longer chamber closure time than recommended (Pavelka et al., 2018) was used to obtain robust Keeling plots along with the flux measurements."

LN 137-138: How long were samples stored in exetainers before being shipped back to Europe for analysis? Some literature suggests that more extended storage periods of gas-filled exetainers before analysis could lead to sample contamination/degradation. Besides the overpressure, what else did you do to lower the risk of sample degeneration/degradation during storage in DRC and during shipping to Europe?

For logistical reasons, it was not always possible to regularly ship the samples back to Europe due to the remote location of the sampling site. The time between collecting and analyzing was often several months. Besides storing the samples at overpressure and in a dark cool place an additional layer of silicone was applied on each vial septum before evacuation. This additional silicone layer helped preserve the vacuum as well as the sample. Given the high quality of our coefficient of determination of the linear relationship between the concentration increase over time we are confident that minimal sample degradation occurred. If that was the case more erratic concentration increases would have been observed frequently. We added the following line to the method section:

'To aid vacuum and sample preservation, each evacuated vial was sealed with an additional silicone layer (Dow Corning 734, Dow Silicones Corporation, USA).'

LN 181-182. Could you add the flush time?

The flushing time is dependent on the gas flow rate. As a rule of thumb, we flushed each vial with at least ten times its own volume of gas. With our standard operational procedure, we usually $N_2$ flush 30 vials (12ml) at a flow rate of 5 L/min for about 3 mins which is sufficiently long to replace residual air in the vial with $N_2$. Flush quality was frequently checked using gas chromatography.

LN 229 What did you do to the data if the model assumptions did not conform to normality and homoscedasticity prescriptions?

The model presented in the manuscript conformed to the assumptions; hence no transformation (except standardization on the predictors) was applied to the data.

LN 235—As Reviewer #1 indicated, please update the figure numbers accordingly. There is a mismatch between the figure numbers and the in-text citation.

Mismatched figure numbers and references in the text have been corrected. Our apologies for this issue with updating properly the references.

LN 252 You measured the fluxes every other fortnight, but you report mean weekly fluxes. This is confusing. Clarify

Thank you for highlighting the confusion regarding the data displayed. To ensure clarity, we have addressed your comments on LN 252 and LN 260 together, providing our response below the comment on LN 260.

LN 260: Could you add vertical lines to the graph showing the beginning and end of the dry season over the three years? I am also slightly confused about how you present weekly fluxes when measurements were done fortnightly. Did you interpolate fluxes for the weeks you did not measure? Could you also add a short note in the figure caption about why some time series are discontinuous?

For better contextualization, vertical bands were added to the graph to represent the typical flooding period (Jan-Dec). We focus on the flooding period (Dec-Jan) rather than the wet/dry season. Defining a wet/dry season is difficult as there are frequent rain events throughout the year (see Figure 3). However, we added in the text which months are considered to be the dry and wet seasons as follows:

'The long dry season in Mbandaka is considered from July to August whereas the short dry season spans between January and February. However, frequent rainfall as shown in Figure 3 renders the region as relatively wet throughout the entire year. Annual precipitation was the highest in 2020 with 1855 mm and lowest in 2022 with 1417 mm (self-measured; Figure 2). The flooding period at the study site is typically centered around December and January.'

Thank you for pointing out the confusion regarding the data displayed. It is currently poorly described. We decided to display a weekly over a fortnightly resolution to maintain a higher resolution of the environmental parameters and to better streamline the data between the two sites which were not always measured during the same week. Moreover, for some periods, more intensive measurements were available (daily, weekly). All $CO_2$ flux data points (cross) presented in Figure 3C correspond to individual fluxes measured, while only the line displayed corresponds to the weekly means linearly connected. To clarify this, as well as the discontinuity, the legend has been modified as follows:

**Figure 2. Weekly precipitation, volumetric soil water content, temperature, and $CO_2$ fluxes. (A) The sum of the weekly precipitation [mm] (blue) obtained from the Trans-African Hydro-Meteorological Monitoring Observatory and mean volumetric soil water content [$m^3$ $m^{-3}$] measured with soil moisture sensors (ECH$_2$O 5TM = solid line, TMS-4 dataloggers = dotted line). (B) Distribution of volumetric soil water content [$m^3$ $m^{-3}$], both sensor types combined. (C) Mean weekly air temperature [°C] (gold) was obtained from the Trans-African Hydro-Meteorological Monitoring Observatory. The mean weekly soil temperature [°C] was measured with soil temperature sensors (ECH$_2$O 5TM = grey solid line, TMS-4 dataloggers = grey dotted line). (D) Distribution of air and soil temperatures [°C], both sensor types combined. (E) Measured surface $CO_2$ fluxes (cross) [$\mu$mol $m^{-2}$ $s^{-1}$] from the SSF site (brown) and calculated $CO_2$ fluxes from the PFF site with a K of 3.5 cm $h^{-1}$(pink). Calculated weekly means (line) and the standard error of the mean are displayed. Blue shading represents river levels (see Section 2.6), while grey bands indicate flooding periods (Dec-Jan) at the SFF site. The displayed time series are discontinuous due to fieldwork constraints (see Section 2.2.) (F) Distribution of surface $CO_2$ fluxes at the PFF and SFF sites.**

LN 286: "Individual relationships between soil CO2 fluxes and the environmental parameters (soil temperature (A), volumetric soil water content (VWC) (B) and river level (C))". Measures taken while the soil chamber was partially flooded are represented in green. Regression lines are displayed in brown. These sentences sound like figure captions and not results. Please reword.

Indeed, our apologies for this mistake. These sentences are the figure caption that had been added by mistake with the Figure cross-reference. This was corrected: the caption has been removed from the main text.

LN287: $CO_2$ and not CO2

The format of $CO_2$ has been corrected.

LN 295: Could you display the r-coefficient values for Figures 4 a-b. The positive correlation displayed in Figure 4 a looks very weak, at least visually, and it would be misleading to add a regression line to this figure if r < 0.5. Also, could you briefly talk about the strength of these correlations when first introducing them in the results section? You should indicate whether the r values are less or greater than 0.5 in the text description. Additionally, for Figures 4 c-d, add the quadratic equations.

Thank you for pointing out this lack of clarity. Figures 4a-d intend to illustrate the relationship between the different explaining variables and the dependent variable of the LMER model presented in Table 1. In that sense, the $R^2$ value to consider is the one related to the complete LMER model (0.354/0.430). However, we agree with the reviewer that this might be misleading. To address this issue, we have modified the plots, so that the line appears as a suggested grey dashed line, clarified the figure caption as written below, and added the inclusive $R^2$ on each plot. The inclusive $R^2$ provides a measure of the proportion of the variance explained by each predictor, including both its direct effects and its interactions with other predictors, whereas part $R^2$ quantifies the unique contribution of each predictor, considering only its direct effects (i.e., without interactions; Stoffel et al., 2021). We suggest this approach as it can provide the reader with a clear way to understand the overall contribution of each predictor and its relationship with the dependent variable. Finally, the method section was modified accordingly to explain how the inclusive $R^2$ was calculated (R package partR2)

The method section for the statistical analyses now reads as follows:

'Marginal and conditional $R^2$ values for mixed effects were calculated using Nakagawa et al., (2017), inclusive $R^2$ estimated with *partR2* package (Stoffel et al., 2021) and p-values using Satterthwaite's approximation with the *lmerTest* package (Kuznetsova et al., 2017).'

The line 286 was also modified to clarify the link between Table 1 and Figure 4, as follows:

'For a more complete understanding of the LMER model ('Table 1), the individual relationships between surface fluxes and the different predictors (soil temperature, river level, and soil moisture) as well as the effect of the interaction between the soil temperature and rivel level are visualized in Figure 3. The inclusive $R^2$ ($IR^2$) of each predictor is also presented, offering a measure of the proportion of variance explained by each predictor, including both its direct effects and interactions with other predictors (Stoffel et al., 2021). In this context, the soil temperature ($IR^2$ = 0.225), soil moisture ($IR^2$ = 0.126), and the quadratic component of the river level ($IR^2$ = 0.097) appear as the primary factors explaining the variance in surface $CO_2$ fluxes, whereas the interaction between soil temperature and river level ($IR^2$ < 0.001), along with the linear component of the river level ($IR^2$ = 0.001), make no meaningful contribution (Figure 3).'

The figure caption was modified as follows:

**Figure 3. Individual relationships between soil $CO_2$ fluxes and the environmental parameters (volumetric soil water content (VWC) (A), soil temperature (B), and river level (C)). Measures taken while the soil chamber was partially flooded are represented in green. Regression lines are displayed as dashed black lines. The interaction between soil temperature and river level is illustrated. Values were predicted based on the LMER model ('Table 1) (D). Inclusive $R^2$ (A, B, C) was estimated based on the LMER model ('Table 1; Stoffel et al., 2021).**

LN 314. Add a full stop at the end of the line. General comment on the results. I miss the discernment between the results that were statistically significant and those that were not. The Authors use phrases like slightly increased, strongly enriched, etcetera, but do not specify whether this was a statistically significant change. Could you add p-values in brackets where you describe changes in measured fluxes and environmental controls?

We added a full stop at the end of the line. Linear models were tested between the aquatic fluxes (PFF) and explaining variables (pH. C:N ratio, river level etc.). None of the linear relationships tested were statistically significant. The dataset, however, was small (n = 28) with high variability in the response. This has been clarified in the text by explicitly mentioning that the relationship was not statistically significant and/or adding the p-value in brackets if applicable. Furthermore, on second thought and based on the reviewers' feedback, we decided to remove Figure 5 from the manuscript as it does not add much value.

The corresponding result section (line 299) has been modified as follows:

'At the PFF site, surface $CO_2$ fluxes did not exhibit statistically significant relationships with pH, river level, carbon to nitrogen ratio (C:N), dissolved organic carbon, or biodegradable dissolved organic carbon. Trends were observed, such as an increase in $CO_2$ fluxes with rising river levels and opposing trends with pH, $CO_2$ fluxes appeared to decrease as pH increased. However, these are just visual tendencies and not statistically significant.'

LN330, why are your fluxes on the lower end? Please substantiate.

This is indeed a very interesting question. Our data falls within the lower range measured across the pan-tropical realm which could have been caused by site-specific characteristics such as local climate, soil type, and/or vegetation. It is challenging to offer a definitive explanation. Among the literature, Baumgartner et al., (2020) and Daelman et al., (2024) measured CO2 fluxes in the Congo Basin. When comparing the lower fluxes observed at our SFF sites with those reported by Baumgartner et al. (2020) in the Congo Basin lowland forests, one plausible explanation is that the flooding period at the SFF site significantly influences the annual mean flux.

To further discuss our results and shed light on the distinctions between sites, we have focused our comparison on the PFF and SFF sites—using other studies primarily to provide context. The following text was added in section 4.1. to compare both sites:

' The elevated $CO_2$ fluxes at the PFF site resulted in higher fluxes relative to the SFF site. Further research is needed to determine whether a greater water depth-integrated respiration (Amaral et al. 2020), a positive correlation with a larger inundated area (Amaral et al. 2020), prolonged river interactions or other factors could explain such difference. In contrast, the SFF site presented reduced $CO_2$ fluxes during the onset of flooding, speculatively due to the inhibitory effect of excessive soil moisture on soil respiration (Courtois et al. 2018; Nissan et al. 2023).'

LN 332 Please reference the other flooded forests you are referring to here and give the magnitude of fluxes measured at those tropical flooded forest sites.

The sentence was modified as follows:

'The perennially flooded forest site (PFF), located at the interface between terrestrial (forest) and aquatic (stream) ecosystems showed relatively high emissions (4.38 ± 0.64 µmol m-2 s-1) when compared to other tropical flooded forests (Scofield et al. 2016; **Error! Reference source not found.**) or those streams draining catchments dominated by seasonally or continually inundated swamp forests (Mann et al., 2014; Alin et al. 2011; **Error! Reference source not found.**).'

LN 336. You need a reference here.

In response to Reviewer 1's comments, this paragraph was modified as follows:

'A non-significant positive trend between water level and the aquatic $CO_2$ fluxes was visually discernible which is in line with a positive relationship between $pCO_2$ and discharge measured on the adjacent Ruki (Drake et al., 2023). As a constant gas transfer velocity was used in the present study, short-term changes in aquatic $CO_2$ fluxes reflect the variations in carbon dioxide concentrations ($pCO_2$) in the water.'

The statement regarding the direct relationship between aquatic $CO_2$ fluxes and dissolved $pCO_2$ is based on the calculation method of this paper and not on an external study. That is, the use of a constant gas transfer velocity in the study to calculate $F_{CO_2}$ implies that variations in $CO_2$ fluxes are directly linked to variations in $pCO_2$.

LN 341 Please add a reference (s)

The text was amended as follows:

'Generally, the $pCO_2$ concentration itself is driven by factors such as terrestrial inputs, gas exchange with the atmosphere, water temperature (gas solubility), water chemistry (pH, alkalinity), and in-stream metabolism (Rocher-Ros et al. 2019; Hotchkiss et al. 2015; Battin et al. 2023).'

LN 325-355: Your arguments need elaboration and should be backed by literature. General comment on the results. Please add a bar graph of the magnitude of the measured fluxes and all the controls showing the differences in these variables between the SFF and PFF sites across the wet and dry seasons. See suggestion below

Section 4.1. was further elaborated and referenced thanks to the previous comments and the feedback from Reviewer 1. The main text additions were written in response to the comments on LN 331 and LN 23. Thank you for detailing your explanations with a drawing. Based on your comment, Figure 3 has been revised to include boxplots alongside each graph (A, B, C). We suggest this approach to keep the different graphs in the same panel, allowing direct comparison of the time series and the fluxes distribution. These boxplots illustrate the distribution and magnitude of the measured fluxes at the SFF and PFF sites, as well as key environmental variables such as precipitation, soil moisture, soil temperature, and air temperature. However, we recommend avoiding the distinction between dry and wet seasons, as frequent rainfall throughout the year makes it challenging to define clear seasonal boundaries and rather suggest indicating on the plot the typical flooding periods (see Figure 3).

[Figure]

LN 378. It is interesting to note the increasing evidence of the lack of seasonality in CO2 fluxes. There have been a couple of studies in the region that have found similar results. It might be necessary to cite them here to back up your findings.

Thank you for pointing this out. We added references to Baumgartner et al. (2020) whose study showed weak seasonality in $CO_2$ fluxes in the lowlands forests of the Congo Basin.

The start of the section 4.1 now reads as follows:

'The surface $CO_2$ flux dataset from the SFF site, measured for three consecutive years, showed intra-seasonal and interannual variability. However, no clear seasonal patterns were observed (Figure 2C). Baumgartner et al. (2020) showed a similar low seasonality in lowland forests of the Congo Basin, attributing it to the limited rainfall variation between dry and wet seasons.'

LN 397. There is a typo. The new sentence should read, "in other words"

The typo has been corrected.

LN 430-434, Be consistent in the way you abbreviate the author names Mbartel versus TW

As co-authors Marijn Bauters and Matti Barthel shared the same initials, it was decided to add a complete name in their case but stay concise and keep the abbreviations for the rest of the authors.

---

## Author Response (AR2)

Dear Authors,

Thank you for thoroughly addressing my previous comments.

However, upon reviewing the revised manuscript with track changes enabled, I noticed that lines 135–140 mention the installation of a total of six chambers at the SFF site. This raised a question regarding the sufficiency of this number in capturing the variability in the measured $CO_2$ fluxes. I would have expected you to establish multiple random replicate plots across different locations at the SFF site and then deploy six chambers within each plot. For instance, if you had implemented four replicate plots, each containing six chambers, this would have resulted in 24 sampling locations, which, in my view, would better account for the expected spatial heterogeneity in $CO_2$ flux measurements.

Therefore, I request you to provide additional details in the methodology section regarding the rationale for sampling gases at only six locations at the SFF site. Additionally, it would be helpful to specify the spatial arrangement of the chambers (distance between chambers)—whether they were closely or widely spaced. If this sampling design presents a limitation to the study, I encourage you to acknowledge and discuss this aspect in the discussion section.

It is also not clear how many locations within the PFF site were sampled and what was the separating distance between the sampling locations.

Aside from this concern, I am satisfied with the improvements you have made in the revised manuscript.

Dear Reviewer,

Thank you for your additional comments.
We generally agree with your opinion that having more sampling points would have been beneficial for better capturing the spatial heterogeneity at the site. Due to the remote location and the resulting logistical constraints, however, we decided to opt for a rather minimalistic setup. Using, as suggested, 24 chambers would have resulted in 96 vials (4 time points times 6 chambers) per sampling day; and managing such a large number of vials from/to Mbandaka was not feasible from an organizational standpoint (vial supply, vial retrieval). To overcome these restrictions, we selected one representative forest site together with local researchers, considering species composition and flood regime. The chambers were then placed accounting for local microtopography. The chamber distance was kept at about 20m in order to be able to sample the

chambers in parallel rather than in sequence. Regarding the PFF site, we assumed that the flooding water was homogenous, thus keeping the replication at a statistical minimum (three replicates). Site homogeneity is reflected in the very low variability among the replicates within one sampling day.

Additionally, we amended the manuscript as follows :

Line 126 : The SFF site was chosen as a representative site of the surrounding forest. The six chambers were spaced about 20 meters apart, randomly distributed across the site but accounting for variations in local microtopography.